# Understanding Medical Time Series Event Piece by Piece: A Fine-Grained Event Detection Network

## Abstract

Event detection in medical time series is fundamental to supporting health monitoring and clinical decision-making. However, most existing methods divide time series into fixed-length segments and perform coarse-grained, segment-level detection, which fails to precisely localize the start and end times of events. This limitation can mislead clinical assessment and obscure the true severity of conditions. To address this, we propose EventCompreNet——a universal network for fine-grained event detection leveraging auxiliary tasks. Inspired by the cognitive processes that human detect events, we introduce four human comprehension tasks to enhance the model's understanding of each piece of events. Moreover, to overcome the limited knowledge transfer in existing multi-task learning structures, we develop a task-deep-coupling framework that facilitates deeper interaction among tasks. Through these designs, EventCompreNet achieves a comprehensive understanding of the entire event life cycle. Experimental results on four benchmark datasets demonstrate that our model significantly outperforms existing state-of-the-art time series models in fine-grained event detection and exhibits strong event comprehension capabilities.

## 1 Introduction

The detection of medical time series events can provide critical guidance for health monitoring and disease diagnosis Levy et al. (2023); Jia et al. (2021); Ahmad et al. (2023). Traditional event detection tasks in time series analysis—such as sleep staging, epilepsy detection and sleep apnea detection—have currently achieved strong performance within their respective domains, but they typically segment signals into **fixed windows** (e.g., 30 s–several minutes) for classification. They fall into the category of **coarse-grained** event detection approaches, which lack the flexibility to lpinpoint event onsets and offsets, limiting both diagnostic accuracy and assessment of disease severity.

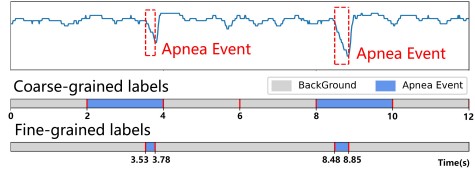 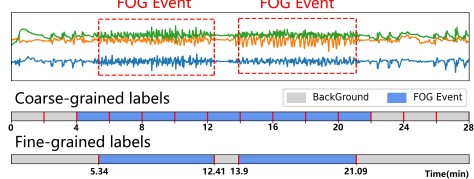

(a) Obstructive sleep apnea detection      (b) Freezing of gait detection in Parkinson's disease

Figure 1: Differences between coarse-grained and fine-grained event detection labels. (a) Compared with coarse-grained labels, fine-grained labels can accurately mark the duration of apnea events, thus be used to assess the severity of a patient's condition. (b) In Parkinson's freeze of gait (FOG) detection task, the coarse-grained labels merge two events into one, making an inaccurate event count.

Thus, a kind of **fine-grained** time series event detection is required. As shown in Fig. 1, unlike coarse-grained tasks, it localizes events at the **frame level** (point level), precisely marking onsets and offsets and enabling accurate event counts. Such precision supports diverse tasks—including sleep spindle detection, Parkinsonian freezing of gait (FOG), obstructive sleep apnea (OSA), and

ECG waveform delineation Kaulen et al. (2022); Bikias et al. (2021); Levy et al. (2023); Urteaga et al. (2025)—providing clinicians with reliable, objective measures of disease severity. However, to the best of our knowledge, few studies address fine-grained event detection, and existing efforts are task-specific with several unresolved challenges:

**How to build a universal fine-grained event detection model is a challenge.** The difficulty arises from two core issues: (1) Fine-grained event detection has high requirements for the localization of the start and end time (event boundaries), while few existing studies address this You et al. (2021); Kaulen et al. (2022). (2) Moreover, event durations vary widely—for example, sleep spindles last 0.5–2 s Iber et al. (2007), while obstructive apnea can span 10–45 s Levy et al. (2023).

The key to solving these problems is enhancing the model's comprehension of the ongoing development status of events. Drawing on insights from human event cognition Zacks & Tversky (2001); Zacks et al. (2007); Iber et al. (2007), we decompose medical events into multi-aspect functional pieces, then implement this decomposition through four targeted learning tasks. We term these Human Comprehension (HC) tasks, which leverage four distinct annotation schemes to guide models in hierarchically assembling event understanding from multiple perspectives, piece by piece.

In the scenario of this study, while many auxiliary tasks carrying a variety of valuable knowledge are stacked on the model, effective knowledge transfer across tasks via multi-task training remains difficult with a simple framework, presenting our second challenge.

**How to effectively transfer knowledge from auxiliary tasks to the model is the second challenge.** Traditional auxiliary task framework employs a task-shared encoder for common features shared by all tasks, while each task has a dedicated decoder to generate its specific output (see Appendix B for more details). This design poses two key issues: (1) Knowledge Transfer Restriction: Valuable knowledge relevant to the main task is captured by auxiliary task-specific decoders but tends to remain isolated within these decoders during the subsequent optimization process, without being effectively transferred to the task-shared area. (2) Model Parameter Increase: Incorporating many task-specific decoders brings a certain load to the amount of parameters of the model, making optimization more challenging and causing redundant parameter usage. To address these, we propose a task-deep-coupling framework to maximize the task-shared area while keeping parameter growth in check.

To sum up, the contributions of this study are as follows: (1) We propose EventCompreNet for fine-grained medical time series event detection. To the best of our knowledge, it is the first universal network in this research area. (2) Referring to human detection and comprehension patterns of events, we introduce four human comprehension tasks to enhance the model's understanding of events from multiple perspectives. (3) We develop a task-deep-coupling framework that improves the inter-task knowledge transfer while avoiding parameter growth caused by adding tasks. (4) Experimental results on four datasets demonstrate that the proposed model achieves state-of-the-art performance. Furthermore, model visualizations illustrate its capability to comprehend events. The source code for the proposed model is provided in the supplementary materials.

## 2 RELATED WORK

Traditional fine-grained medical time series event detection methods commonly rely on thresholding techniques Yücelbaş et al. (2018); Lacourse et al. (2019b). They tailor the filter range based on time and frequency domain characteristics. These methods perform well for simple events, but require manual design of filtering schemes tailored to different events, which becomes challenging for complex events. The emergence of traditional machine learning methods Hekmatmanesh et al. (2017) has provided new technical means for detecting complex time series events, but they still rely on the manual design of features. With the prevalence of deep learning methods, models are now capable of automatically extracting complex features. However, deep learning-based approaches for fine-grained event detection in this domain remain limited. Most existing studies have focused on specific tasks such as sleep spindle detection and ECG waveform identification. For sleep spindle wave detection, SpindleNet Kulkarni et al. (2019) uses the CNN+RNN structure to implement an online spindle detection model. It has high efficiency and good detection accuracy. SpindleU-Net You et al. (2021) introduces a U-Net structure network for sleep spindle wave detection. It develops an attention module to focus on the salient spindle region and designs a new loss function for the

class imbalance problem. SUMO Kaulen et al. (2022) is another sleep spindle detection network based on U-Net structure. Its detection results are more similar to the expert-labeled results and have lower model complexity. For ECG waveforms detection, recent works such as Urteaga et al. (2025); Wang et al. (2023); Liu et al. (2021) propose deep learning-based methods for precise delineation of characteristic waveforms include P wave, QRS wave and T wave, showing strong performance on clinically challenging cardiac events. Other medical time series event detection tasks largely remain at the coarse-grained detection stage Levy et al. (2023); Hu et al. (2023); Tang et al. (2022); Liu et al. (2024).

In general, many valuable fine-grained detection tasks have not been fully explored in medical time series. Moreover, few models provide solutions for event boundary localization or event duration adaptation. In addition, there is less research yet on constructing a universal framework for fine-grained medical time series event detection.

# 3 PRELIMINARIES

**Fine-grained time series event detection:** The fine-grained time series event detection task can be defined: Providing an input time series $X = \{x_1, x_2, \ldots, x_L\}$ with a fixed window length $L$, judge the event class for each frame of $X$ as $Y = \{y_1, y_2, \ldots, y_L\}$. The label in a time frame $t$ is $y_t \in \{0, 1, \ldots, k\}$, where $\{1, 2, \ldots, k\}$ denotes the class of events and number 0 means the background class. Assuming no overlap between events, each $y_t$ has a single class.

**Multi-task model:** The multi-task model can be defined: Given an input time series $X$ and a task identity $m$ (task ID), predict the corresponding task result $Y_m$.

# 4 MULTI-PERSPECTIVE COMPREHENSION NETWORK

We propose EventCompreNet for fine-grained medical time series event detection as shown in Figure 2. The backbone of the EventCompreNet is designed according to the U-Net structure Ronneberger et al. (2015). With different inputs of task IDs, the network can output four kinds of HC tasks and a fine-grained time series event detection task (main task), respectively.

We summarize 4 key ideas of the network: (1) Four kinds of HC tasks are designed to improve the depth of the model's understanding of time series events. (2) A task-deep-coupling framework is developed to maximize the degree of information interaction between tasks by maximizing the task-shared area. (3) The coarse-grained event perception task in HC tasks is also used to filter background sequences, which provides a relative class-balanced environment for detection. (4) The proposed network no longer designs specific layers for different tasks. This reduces the complexity of the network and makes it easier to optimize.

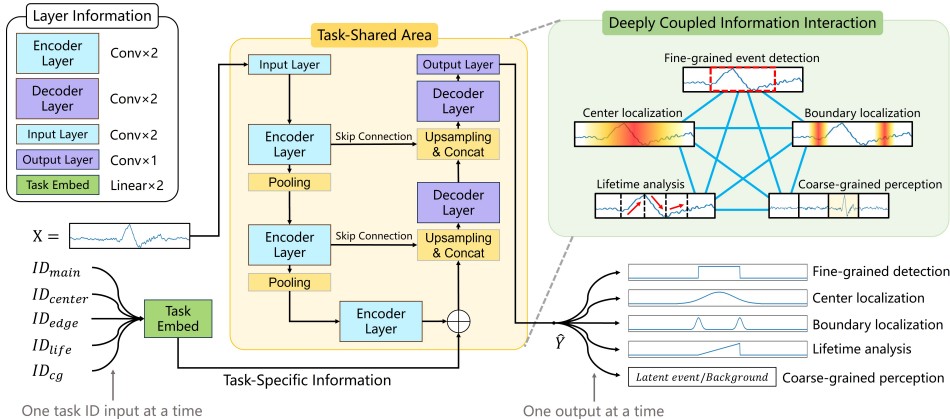

Figure 2: The main structure of EventCompreNet. The model is built on the U-Net structure. It takes time series signals as input and uses task IDs as a switch to control the output task. By maximizing the task-shared area, 4 human comprehension tasks and main task get deep information interaction, which makes the model possess a deep comprehension of the events from different perspectives.

## 4.1 MULTI-PERSPECTIVE HUMAN COMPREHENSION TASKS

By breaking down the process of human understanding time series events, as shown in Figure 3, we propose 4 human comprehension tasks from different perspectives that guide the model to increase its comprehension of the event piece by piece. The four HC tasks are coarse-grained event perception, center localization, boundary localization, and lifetime analysis. Train on these four tasks improves the ability of the model to detect the presence, center, boundaries, and life cycle of the event, which makes the model understand the event from multiple perspectives. These HC tasks are eventually represented by their specific labels, which can be generated from the fine-grained event detection label. The model outputs the predicted labels for these tasks, thus providing an explicit interface for measuring and optimizing the performance of these HC tasks. This allows us to optimize all tasks using supervised learning. Including the fine-grained event detection task, the purpose and construction details of these tasks are described below. In order to facilitate the description, the following introduction is described from a binary classification perspective (classify one event and the background).

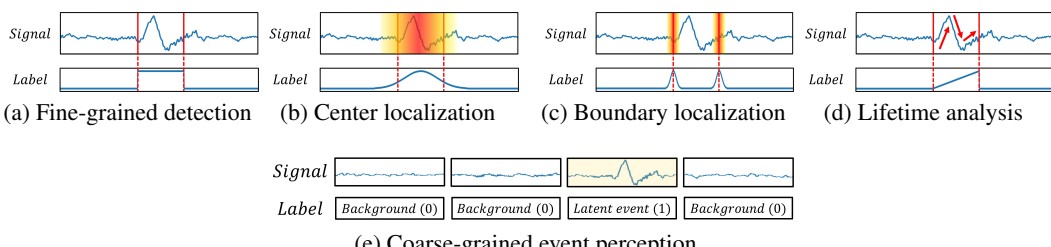

(a) Fine-grained detection  (b) Center localization  (c) Boundary localization  (d) Lifetime analysis

(e) Coarse-grained event perception

Figure 3: The multi-perspective human comprehension tasks with their labels. Figure (a) is the origin fine-grained time series event detection task, plotted here to compare with other tasks.

**Fine-Grained Event Detection Task (Main Task):** As shown in Figure 3a, the mission of fine-grained event detection is to detect the event at frame-level. From a binary classification perspective, the event detection label with input window length $L$ is represented as:

$$Y_{main} = \{y_1, y_2, \ldots, y_L\}, \quad y_i \in \{0, 1\} \tag{1}$$

where $Y_{main}$ means the label series of main task, and $y_i$ ($i \in \{0, 1, \ldots, L\}$) denotes the $i$-th time frame of label. In the background frame $y_i$ is marked as number $0$ and in the event frame $y_i$ is marked as number $1$.

**Center Localization Task:** As shown in Figure 3b, the target of the center localization task is to locate the approximate center of the event and prepare for finer detection. It helps the model to focus on the core region for event detection.

The Gaussian distribution function is suitable for gradually marking the center of the event Dai et al. (2022). The closer to the center the higher its activation. Given $S = \{t_1, t_2, \ldots, t_L\}$ as the timeline of input, the marking process can be defined as:

$$y_i = \text{Gaussian}\,(t_i, \mu; \sigma) = \frac{1}{\sqrt{2\pi}\sigma} \exp^{-\frac{(t_i - \mu)^2}{2\sigma^2}}, \tag{2}$$

$$\text{G}\,(S, \mu; \sigma) = \{y_1, y_2, \ldots, y_L\}, \tag{3}$$

where $\text{Gaussian}(\cdot, \mu; \sigma)$ and $\text{G}\,(\cdot, \mu; \sigma)$ are the frame-level and sequence-level Gaussian distribution generating function, respectively. Given $N$ as the event number in the input signal, $t_{mid}^e$ and $d^e$ as the center time and duration of the $e$-th event, the label of center localization task $Y_{center}$ is generated as:

$$H_{center}^e = \text{G}\,(S, t_{mid}^e; d^e/2), \quad e = 1, 2, \ldots, N, \tag{4}$$

$$Y_{center} = \max_{e=1,2,\ldots,N} H_{center}^e, \tag{5}$$

where $H_{center}^e$ is the center label of event $e$. $\mu$ and $\sigma$ in Gaussian distribution function is chosen as $t_{mid}^e$ and $d^e/2$, respectively. When multiple event labels overlap at the same time point, the maximum value is taken.

**Boundary Localization Task:** As shown in Figure 3c, the boundary localization trains the model to concentrate on the signs of the start and end of the event. The ideal boundary label should only have two labels at the event start frame and end frame, which makes it difficult for the model to mine valuable information. Therefore, in order to gradually guide the model to discover the harbingers of event occurrence and termination in context, we use the Gaussian distribution function to make a progressive activation curve centered on the boundaries:

$$H_a^e = \mathrm{G}\left(S, t_a^e; d^e/6\right), \quad a = start, end \tag{6}$$

$$Y_{boundary} = \max_{\substack{a=start,end \\ e=1,2,\ldots,N}} H_a^e, \tag{7}$$

where $t_{start}^e$ and $t_{end}^e$ are the start and end time of event $e$. $H_{start}^e$ and $H_{end}^e$ are the boundary labels of the $e$-th event. $\sigma$ in the Gaussian distribution function is chosen as $d^e/6$. $Y_{boundary}$ is the boundary localization label. The closer to the boundaries, the higher the activation of the label.

**Lifetime Analysis Task:** As shown in Figure 3d, the target of the lifetime analysis task is to infer the progress of events at each frame, which makes it the most challenging task of all HC tasks. This task is used to help the model understand all stages of the event development process. Thereby, the model has a complete comprehension of the event life cycle.

An interval from 0 to 1 is used to map the percentage process of events. Given the event duration $d$, the event lifetime label is generated as:

$$y_i^e = \begin{cases} 0 & , t_i \notin [t_{start}^e, t_{end}^e] \\ \dfrac{t_i - t_{start}^e}{d} & , t_i \in [t_{start}^e, t_{end}^e] \end{cases}, \tag{8}$$

$$H_{life}^e = \{y_1^e, y_2^e, \ldots, y_L^e\}, \tag{9}$$

$$Y_{life} = \max_{e=1,2,\ldots,N} H_{life}^e, \tag{10}$$

where $H_{life}^e$ is the lifetime label of event $e$, and $Y_{life}$ is the integrated lifetime label of input signal.

**Coarse-Grained Event Perception Task:** As shown in Figure 3e, the target of the coarse-grained event perception task is to be aware of whether there are latent events in the segment level input sequences. Fine-grained event detection usually has a stronger class imbalance with a huge amount of background class. This task helps the model learn features that can be used to exclude background data and help with class balance. The label of coarse-grained event perception task is denoted as:

$$Y_{cg} = \begin{cases} 0, & \max\left(Y_{main}\right) = 0 \\ 1, & \max\left(Y_{main}\right) = 1 \end{cases}, \tag{11}$$

where $Y_{cg}$ is the label of coarse-grained event perception task, and $\max\left(\cdot\right)$ chooses the max number in series $Y_{main}$.

## 4.2 Task-Deep-Coupling Framework

As shown in Figure 4a, the task-shared area of the traditional multi-task framework has certain limitations. This inhibits the space for information interaction and causes knowledge transfer restrictions on task-specific decoders. Therefore, in order to solve the above problems, we propose a task-deep-coupling framework (TDC framework). The following describes its implementation.

**Task-Shared Area Maximization:** As shown in Figure 4b, in the TDC framework, the decoder is changed from a task-specific layer to a task-shared layer with a task ID switch to control the output task type. This makes the decoder a new task information interaction area. In addition, it avoids the knowledge transfer restriction through parameter sharing. Eventually, all data processing layers (the encoder and decoder) of the model are shared by all tasks to maximize information combination space. The learning process of the model on different tasks is deeply coupled. Given input sequence $X$ and task number $ID_{task}$, the process of TDC framework can be defined as:

$$H = \mathrm{Encoder}\left(X\right), \tag{12}$$

$$T_{task} = \mathrm{Embed}\left(ID_{task}\right), \tag{13}$$

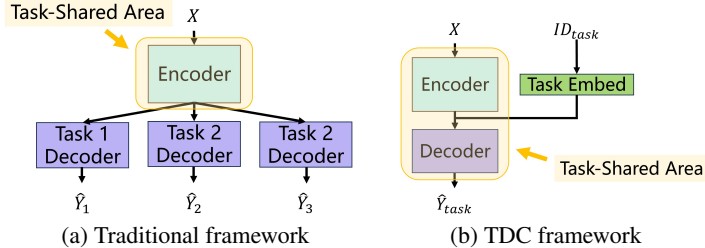

(a) Traditional framework        (b) TDC framework

Figure 4: Traditional multi-task framework and task-deep-coupling framework. (a) Traditional framework have task-specific decoders, which bring certain limitations to information exchange. (b) Task-Deep-Coupling (TDC) framework maximizes the task-shared area and deeply couples the learning processes of different tasks.

$$\hat{Y}_{task} = \text{Decoder}\left(H + T_{task}\right), \tag{14}$$

where $\text{Encoder}$ and $\text{Decoder}$ are the encode and decode function, respectively. $\text{Embed}$ is the task embed operation. $H$ is the encoded feature map, $T_{task}$ is the embedding of $ID_{task}$. The $H$ and $T_{task}$ are fused by point-wise addition and fed into the decoder to get task result $\hat{Y}_{task}$. The TDC framework no longer designs task-specific decoders, thereby avoiding a number of parameters brought by auxiliary tasks.

The task-shared decoder only provides a sequence output with length $L$. Therefore, for the coarse-grained perception task, we apply a max pooling function with kernel size $L$ to transform the decoder output without extra parameter cost.

**Task Synchronous Training:** Simultaneous training of multiple tasks is the most important way to enhance information interaction between tasks. Based on the TDC structure mentioned above, we propose a *task as sample* view to transfer multi-task data and labels for synchronous training: Both task ID and input signal are combined into a new sample, and the new label is corresponding to the original task label. These can be defined as: $X_{new} = \{X, ID_{task}\}$, $Y_{new} = Y_{task}$, where $task = \{0, 1, \ldots, k\}$ represents the number of task, $X_{new}$ and $Y_{new}$ are the new sample and new label. Assuming that the original sample size is $r$, for a scenario with $k$ tasks, the process will provide $r \cdot k$ samples. In this way, traditional multi-task learning patterns are converted into a multi-class sample learning pattern. We shuffle the complete set of new samples so that multiple task types appear in one training batch. This allows the model to optimize multiple tasks in one optimization process, thereby exploring complementary information between tasks.

In addition, according to different tasks in a batch, we use the corresponding loss function to calculate the loss and attach the weight of the task. Specifically, the main task and coarse-grained perception task use cross-entropy loss, and other tasks use mean squared error (MSE) loss.

### 4.3 CLASS-BALANCED FILTERING

The proposed method use coarse-grained event perception task as a class-balanced filter before event detection (The data flow is shown in Appendix C). First, the coarse-grained event perception task works on segment level sequences to initially filter the input sequences that are unlikely to have events. In this way, a class-imbalanced environment with a large number of background classes is transformed into into a relatively class-balanced environment. Ideally, each input sequence in the relative class-balanced environment contains at least one event. Next, the data in the relative class-balance environment is used for further fine-grained event identification.

Due to the different data environments in the prediction period, it is necessary to change the training environment for each task. Specifically, the coarse-grained perception task is trained on all data, while the other tasks are trained using all the sequences with events. The sample size for each task is repeatedly sampled to align with the largest sample size task.

### 4.4 NETWORK BACKBONE

As shown in Figure 2, the body of the network is built according to U-Net Ronneberger et al. (2015), thus ensuring the same length of input and output sequences. There are 3 Encoder layers and 2

Decoder layers which use double 1D convolutions. The input layer consists of 2 convolution layers for expanding the channels and extracting preliminary features. The output layer uses one convolution layer with kernel size 1 to map the channel to the number of event classes. The maximum pooling layer is used for the downsampling operation, while the deconvolution layer is used for the upsampling operation. Except for the output layer, each convolution layer is followed by a batch normalization layer and the ReLU activation function. In addition, the task embedding layer consists of 2 linear layers.

## 5 EXPERIMENT

We evaluate the performance of proposed model on four fine-grained event detection tasks with different signal types and event durations: FOG detection, sleep spindle wave detection, OSA detection, and QRS complexes wave detection.

### 5.1 DATASETS AND PREPROCESSING

**TDCS-FOG Dataset [1]:** It collects lower-back 3-channel 128 Hz acceleration data from 62 subjects and labels the FOG event of three freezing types (Start Hesitation, Turn, and Walking).

**DREAMS Dataset Devuyst et al. (2006):** It collects 30 minutes sleep electroencephalogram (EEG) data fragment from the 6 subjects which contain sleep spindle wave labels.

**SHHS dataset Zhang et al. (2018); Quan et al. (1997):** It contains obstructive apnea labels of $SpO_2$ signals from 5793 subjects sleeping. The first 1000 subjects are selected for OSA detection.

**QT dataset Urteaga et al. (2025):** An ECG recording Dataset with 112,497 QRS complexes annotated by experts. 71 subjects are selected in our experiment. The details of these datasets are given in Appendix D.

### 5.2 BASELINE METHODS

We adapted three specific medical event detection methods and evaluated them across all experimental tasks: sleep spindle wave detectors **SpindleU-Net** You et al. (2021) and **SUMO** Kaulen et al. (2022), as well as a latest QRS wave detection model **QRSU-Net** Urteaga et al. (2025), published in 2025. Meanwhile, a hand-crafted spindle wave detector called **A7** Lacourse et al. (2019a) is used to compare with data-driven methods. Besides, in order to make up for the lack of technical perspective in fine-grained medical time series event detection, we introduce an effective model in the field of video action (event) detection: **MS-TCT** Dai et al. (2022). Meanwhile, some time series universal feature extraction models are also selected as the baselines: **TimesNet** Wu et al. (2023); **Informer** Zhou et al. (2021); **Non-stationary Transformer (NSTransformer)** Liu et al. (2022); **TimeMixer** Wang et al. (2024), an advanced baseline for general time series modeling. The details of these baseline methods are given in Appendix E.

### 5.3 EXPERIMENT SETTINGS

**Training and Hyperparameter Settings:** We implement the proposed model based on PyTorch framework. The Adam optimizer is used to train the model. On the {FOG, DREAMS, SHHS, QT} datasets, the learning rate is set to $\{10^{-3}, 10^{-3}, 10^{-4}, 10^{-3}\}$, and the batch size is set to {512, 128, 256, 32}. All data are divided into input sequences using non-overlapping sliding windows. The input sequence lengths of the models on the four datasets are {320, 3840, 320, 256}. The model is trained for 150 epochs with a early-stop patience of 20. We use 5-fold cross-validation to evaluate all method. In each fold, 20% of the data is the test set, and the remaining 80% of the data is divided into the training and validation sets by $8 : 2$. Other experiment and hyperparameter settings including the baseline models are detailed in Appendix M.

**Evaluation Metrics:** We evaluate the results of the model from both event and point levels, and report them as the mean and standard deviation over 5-fold cross-validation. **(1) Event-level evaluation:**

---

[1]TDCS-FOG dataset was acquired at `https://www.kaggle.com/competitions/tlvmc-parkinsons-freezing-gait-prediction/data`.

Event downstream measurements such as event start and end time positioning, and duration measurement all rely on event-level detection, making event-level metrics very important. In event-level evaluation, the consecutive positive points in prediction sequence are combined into one event. Referring to previous work Kaulen et al. (2022); You et al. (2021), we use event-level F1-score to measure the performance of the model (See Appendix F for more details). **(2) Point-level evaluation:** Each point is evaluated as an independent classification task. We use F1-score as the point-level metrics. When calculating multiple classes, the metrics of all classes (except background) are averaged for the final result.

## 5.4 COMPARISON WITH OTHER BASELINES

Table 1 shows the results of EventCompreNet and other baseline methods on four fine-grained event detection datasets. The proposed model achieves state-of-the-art results compared with other baselines. In addition, event-level F1-score is the most valuable metric, and EventCompreNet has outstanding event-level detection performance.

Table 1: Performance comparison with baseline methods on four fine-grained medical time series event detection tasks. Event-F1 and Point-F1 means the event-level and point-level F1-Score, respectively. A7 is a hand-crafted detector which can only be used for spindle wave detection.

| Method | TDCS-FOG Dataset | | DREAMS Dataset | | SHHS Dataset | | QT Dataset | |
|---|---|---|---|---|---|---|---|---|
| | Event-F1 | Point-F1 | Event-F1 | Point-F1 | Event-F1 | Point-F1 | Event-F1 | Point-F1 |
| A7 | $-$ | $-$ | $0.262 \pm 0.090$ | $0.361 \pm 0.064$ | $-$ | $-$ | $-$ | $-$ |
| TimesNet | $0.391 \pm 0.012$ | $0.441 \pm 0.011$ | $0.186 \pm 0.101$ | $0.337 \pm 0.063$ | $0.178 \pm 0.019$ | $0.287 \pm 0.013$ | $0.821 \pm 0.082$ | $0.804 \pm 0.053$ |
| Informer | $0.431 \pm 0.016$ | $0.502 \pm 0.023$ | $0.103 \pm 0.115$ | $0.191 \pm 0.098$ | $0.163 \pm 0.006$ | $0.234 \pm 0.006$ | $0.857 \pm 0.030$ | $0.830 \pm 0.008$ |
| NSTransformer | $0.250 \pm 0.011$ | $0.422 \pm 0.019$ | $0.147 \pm 0.149$ | $0.235 \pm 0.119$ | $0.168 \pm 0.045$ | $0.239 \pm 0.014$ | $0.778 \pm 0.083$ | $0.769 \pm 0.066$ |
| TimeMixer | $\underline{0.455} \pm 0.019$ | $0.534 \pm 0.015$ | $0.228 \pm 0.030$ | $0.447 \pm 0.034$ | $0.073 \pm 0.017$ | $0.206 \pm 0.015$ | $0.654 \pm 0.023$ | $0.830 \pm 0.010$ |
| MS-TCT | $0.449 \pm 0.008$ | $0.539 \pm 0.005$ | $0.297 \pm 0.038$ | $0.449 \pm 0.024$ | $0.273 \pm 0.011$ | $0.353 \pm 0.008$ | $0.769 \pm 0.091$ | $0.759 \pm 0.065$ |
| SUMO | $0.422 \pm 0.014$ | $0.614 \pm 0.013$ | $0.402 \pm 0.005$ | $0.543 \pm 0.036$ | $0.285 \pm 0.049$ | $0.366 \pm 0.031$ | $0.893 \pm 0.042$ | $\underline{0.857} \pm 0.021$ |
| SpindleU-Net | $0.432 \pm 0.012$ | $\underline{0.657} \pm 0.005$ | $0.444 \pm 0.051$ | $0.577 \pm 0.042$ | $0.285 \pm 0.042$ | $\underline{0.375} \pm 0.030$ | $\underline{0.923} \pm 0.032$ | $0.829 \pm 0.018$ |
| QRSU-Net | $0.450 \pm 0.017$ | $0.622 \pm 0.022$ | $\underline{0.557} \pm 0.035$ | $\underline{0.608} \pm 0.029$ | $\underline{0.291} \pm 0.023$ | $\underline{0.375} \pm 0.023$ | $0.904 \pm 0.021$ | $0.854 \pm 0.010$ |
| **EventCompreNet** | $\mathbf{0.466} \pm 0.024$ | $\mathbf{0.668} \pm 0.026$ | $\mathbf{0.616} \pm 0.058$ | $\mathbf{0.610} \pm 0.051$ | $\mathbf{0.310} \pm 0.011$ | $\mathbf{0.389} \pm 0.011$ | $\mathbf{0.946} \pm 0.012$ | $\mathbf{0.901} \pm 0.008$ |

Compared to deep learning methods, A7 uses hand-crafted features, which limits its performance in sleep spindle wave detection tasks. Among the time series universal baselines, Informer achieves good event-level detection performance on the FOG task, but shows mediocre results on the other two datasets. Neither TimesNet nor NSTransformer achieves outstanding results in fine-grained event detection tasks. Although these models incorporate advanced feature extraction components such as TimesBlock and Transformer, their overall frameworks are not well suited for fine-grained event detection. TimeMixer exhibits second-best event-F1 on the TDCS dataset, ranking just behind our model and outperforming other universal baselines. Even so, it still demonstrates noticeable limitations regarding event-level detection of certain fine-grained events, such as OSA events in the SHHS dataset. In contrast, MS-TCT can effectively fuse multi-scale features for fine-grained event detection and attains a secondary event-level F1-score on the FOG task. However, due to the differences in data characteristics between video action detection and medical time series event detection, its performance on other datasets remains moderate. SpindleU-Net and SUMO, adopting convolution-based U-Net structures, are specifically designed for sleep spindle detection. They have demonstrated excellent performance on other fine-grained detection tasks. Similarly, QRSU-Net, which is tailored for ECG QRS wave detection, also shows reliable event-capturing capabilities in fine-grained detection settings. These results suggest that although powerful feature extraction components like Transformers are worth attention, it is even more important to design frameworks (like U-Net) that are inherently suitable for fine-grained event detection.

The proposed EventCompreNet employs a task-deep-coupling framework that efficiently transfers knowledge from multiple HC tasks into the model. This enables a comprehensive understanding of event existence, centers, boundaries, and life cycles. As a result, EventCompreNet achieves state-of-the-art performance across four different tasks, making it a highly valuable and universal model for fine-grained medical time series event detection.

In addition, Appendix G also includes experiments on class-balanced filter performance and Appendix H includes the model's performance under different IoU thresholds.

## 5.5 Ablation Study

As shown in Table 2, we perform an ablation study on the proposed HC tasks and the TDC framework on the DREAMS dataset. Four ablation frameworks are considered: **(1) Basic**: Model without auxiliary tasks. **(2) Traditional**: Traditional auxiliary task framework, each task has its specific decoder. **(3) TDC**: Task-Deep-Coupling framework. **(4) TDC + Balanced Filtering**: Based on the TDC framework, the coarse-grained perception task is used to filter the background sequence.

Table 2: Ablation study on the DREAMS dataset.

| Framework | Tag | Coarse | Center | Boundary | Lifetime | Event-F1 |
|---|---|---|---|---|---|---|
| Basic | (a) | | | | | $0.565 \pm 0.025$ |
| Traditional | (b) | ✓ | | | | $0.560 \pm 0.055$ |
| | (c) | | ✓ | | | $0.538 \pm 0.077$ |
| | (d) | | | ✓ | | $0.574 \pm 0.048$ |
| | (e) | | | | ✓ | $0.581 \pm 0.048$ |
| | (f) | ✓ | ✓ | ✓ | ✓ | $0.572 \pm 0.041$ |
| TDC | (g) | ✓ | | | | $0.572 \pm 0.030$ |
| | (h) | | ✓ | | | $0.570 \pm 0.054$ |
| | (i) | | | ✓ | | $0.576 \pm 0.014$ |
| | (j) | | | | ✓ | $0.582 \pm 0.041$ |
| | (k) | ✓ | ✓ | ✓ | ✓ | $0.601 \pm 0.042$ |
| TDC + Balanced Filtering | (l) | ✓ | ✓ | ✓ | ✓ | $0.616 \pm 0.058$ |

The Coarse, Center, Boundary, and Lifetime are the proposed HC tasks. Compared with the non-auxiliary task model (a), not all auxiliary tasks work well on the traditional auxiliary task framework (b–f). When using the TDC framework (g–k), all HC tasks bring positive improvements to the basic model (a) and are better than the traditional framework (b–f). This demonstrates the effectiveness of the TDC framework in improving inter-task information interaction. When class-balanced filtering is added (l), the performance of the model is further improved. This shows that the proposed model can create an effective relative class-balanced environment to improve model performance. In addition, the TDC framework does not generate new parameters as the task number increases (see Appendix I), thus effectively controlling the complexity of the model.

## 5.6 Visualization of Comprehension Tasks

As shown in Figure 5, we visualize HC task outputs sleep spindle wave and FOG detection. The model correctly labels the center, boundaries and process of the spindle wave, yielding precise fine-grained event detections and demonstrating that it effectively learns the knowledge injected by the HC tasks. In contrast, since the main task integrates the knowledge of all HC tasks, when some HC tasks are not well mastered, it may also slightly affect the main task. As shown in Figure 5a, when the lifetime prediction result remains active slightly after the event ends, the end time judged by the main task is also delayed from ground truth. It may reflect a minor limitation of our approach.

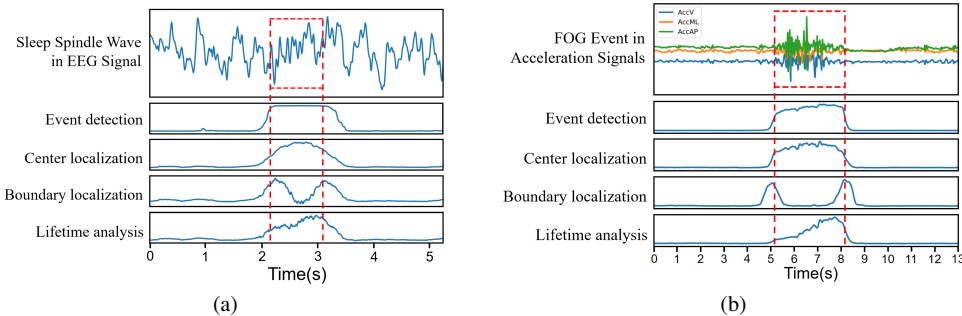

(a)                    (b)

Figure 5: Visualization of EventCompreNet outputs on spindle wave and FOG detection. The model can identify the center, boundaries, and lifetime of events to a certain extent.

## 6 Conclusion

This study proposes a universal framework for fine-grained medical time series event detection. Inspired by the process by that human comprehend events, we design four human comprehension tasks to enhance the model's comprehension of events from multiple perspectives. Meanwhile, we develop a task-deep-coupling framework to enhance knowledge interaction between tasks. The experiments show the proposed model achieves a state-of-the-art performance. In addition, the proposed method can also provide new ideas for other time series tasks.

## ETHICS STATEMENT

This study adheres to the ICLR Code of Ethics and to all relevant regulations for research integrity, data privacy, and human-subject protection. All datasets used are publicly released with documented institutional ethics approvals, and no new data were collected from human subjects by the authors.

### DATA SOURCES AND REPRESENTATIVENESS

We employ four publicly available physiological time-series datasets—TDCS-FOG, DREAMS, SHHS, and QT—covering gait, EEG, SpO$_2$, and ECG signals respectively. Each dataset provides either formal IRB approval or equivalent documentation. A statistical summary of demographic and signal characteristics, along with a bias analysis, is provided in Table 3 to ensure transparency.

Table 3: Statistical summary of datasets used in this study.

| Dataset | Summary |
|---------|---------|
| TDCS-FOG | Subjects: 62, Avg. Age: 69.37, Age Range: 51–94, Gender (M:F): 70%:30%, Race: —, Total Time: 15.3 h, Event Count: 1166, Avg. Event Duration: 17.51 s, Duration Range: 0.18–581.98 s, Sampling: 128 Hz, Signal: Gait (Wearable), Public: Yes (Kaggle), Ethics: Yes |
| DREAMS | Subjects: 6, Avg. Age: 45.67, Age Range: 31–53, Gender (M:F): 50%:50%, Race: —, Total Time: 3.0 h, Event Count: 538, Avg. Event Duration: 0.98 s, Duration Range: 0.39–1.80 s, Sampling: 256 Hz (upsampled), Signal: EEG, Public: Yes (Zenodo), Ethics: Yes |
| SHHS | Subjects: 997, Avg. Age: 57.41, Age Range: 39–89, Gender (M:F): 50%:50%, Race (White:Black): 87%:13%, Total Time: 8132.9 h, Event Count: 26053, Avg. Event Duration: 25.49 s, Duration Range: 2–185 s, Sampling: 1 Hz, Signal: SpO$_2$, Public: Yes (NSRR), Ethics: Yes |
| QT | Subjects: 71, Demographics: —, Total Time: 146.5 h, Event Count: 2156, Avg. Event Duration: 27.64 s, Duration Range: 12–67 s, Sampling: 250 Hz, Signal: ECG, Public: Yes (PhysioNet), Ethics: Yes |

Most datasets focus on adult and elderly populations, consistent with their clinical context (e.g., Parkinson's disease, sleep apnea). DREAMS and SHHS are gender-balanced, while TDCS-FOG shows a male skew consistent with epidemiological prevalence. Only SHHS provides race metadata, limiting subgroup fairness analysis.

### PRIVACY, SECURITY, AND COMPLIANCE

All datasets are de-identified and distributed under their original open-data licenses (Kaggle, Zenodo, NSRR, PhysioNet). No personally identifiable information is accessible to the authors. Data handling complies with HIPAA/GDPR where applicable.

### POTENTIAL SOCIETAL IMPACT

Our methods substantially advance micro-event detection in physiological signals, providing a strong foundation for clinical research and future diagnostic tools, with significant potential to accelerate clinical research and inform next-generation diagnostic systems. While they are not yet intended for direct clinical decision-making, we proactively address potential algorithmic bias and will release code and model weights to enable independent verification and continued fairness evaluation.

In summary, this work complies with ethical guidelines, documents dataset characteristics and limitations, and reflects our commitment to fairness, privacy, and responsible research practice.

## REPRODUCIBILITY STATEMENT

While the full code will be released with the final version, we have already provided the core implementation as described in Appendix K. Moreover, comprehensive model descriptions, along with thorough training and evaluation schemes, are provided to ensure faithful reproduction. Parameter choices and tuning ranges for both our method and baselines are documented in Appendix L, Appendix M, and Appendix N. Together with the publicly available datasets, these resources are sufficient for independent researchers to replicate our methodology and results once the paper is published.

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

## A  THE KEY POINTS THAT HUMANS DETECT AND COMPREHEND EVENTS

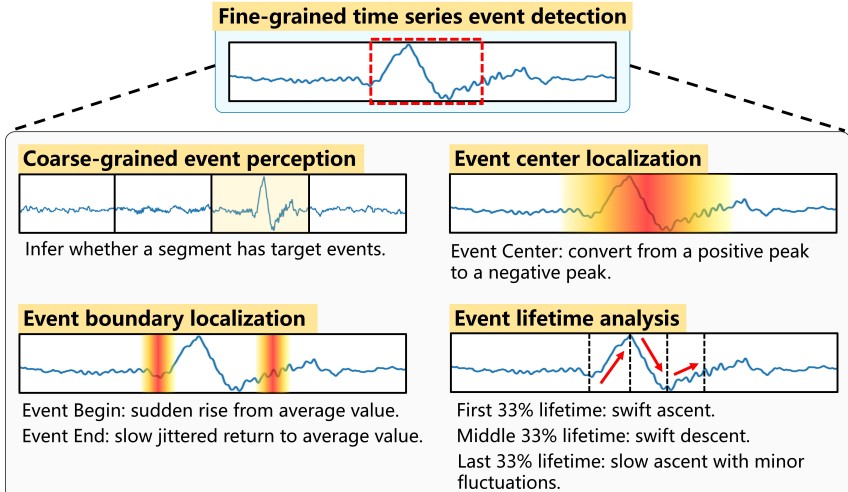

Figure 6: The main job for humans to detect and comprehend time series events. The process of comprehension includes perceiving the existence of an event at a coarse-grained level, learning the center and boundaries of it, and analyzing the development stage of the event throughout its lifetime.

Our framework is grounded in the cognitive strategies that clinical physicians employ when interpreting long-term PSG recordings, rather than merely imitating everyday sensory processing. This perspective motivates the design of our four HC tasks, each reflecting a key stage of human event understanding.

- **Coarse-grained perception:** Physicians scan the night's recordings to filter background and focus on likely event segments—corresponding to our coarse-grained perception task, similar to discarding background frames in video action localization.
- **Event localization (Center & Boundary):** In these segments, physicians identify the core region of an event (analogous to the action climax). The center task infers the event extent from the center outward. The boundary task determines start/end times, complementing the center task by refining event boundaries "outside in."
- **Event integrity judgment (Lifetime):** Physicians also assess event completeness, corresponding to our lifetime task, which models event continuity.

These processes are multi-view and interactive, not strictly sequential. The four HC tasks mirror these perspectives—progressing coarse-to-fine, inside-out, and outside-in, while modeling event integrity. References Iber et al. (2007); Zacks et al. (2007); Zacks & Tversky (2001) on event perception and event structure theories also inspired our task design, highlighting that humans perceive events through dynamic, hierarchical segmentation. This provides crucial cognitive-theoretical inspiration for the design of HC tasks.

## B  THE KNOWLEDGE TRANSFER RESTRICTION OF TRADITIONAL AUXILIARY TASK FRAMEWORK

Knowledge Transfer Restriction: As shown in Figure 7, the task-shared area pertains to the data processing layer that is common to all tasks. It serves as a crucial space for the model to uncover and integrate relevant knowledge across tasks. Typically, optimization enhances the performance of auxiliary tasks, while it does not necessarily improve the performance of the primary task. One possible reason is that the valuable knowledge for the main task is learned by auxiliary task-specific decoders and is continuously retained in these decoders during training, without being effectively transferred to the task-shared area.

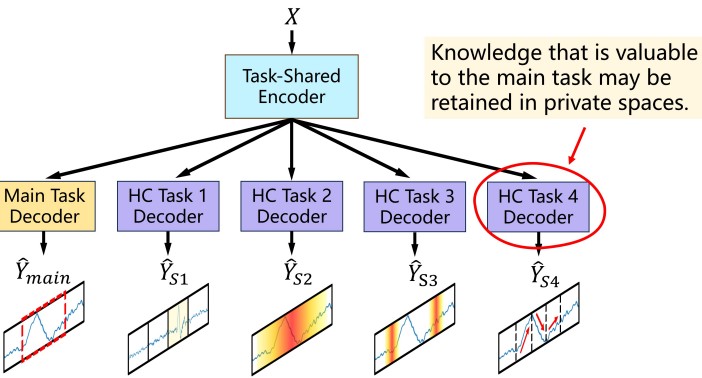

Figure 7: Traditional auxiliary task framework. The proposed HC tasks act as examples. The model is expected to learn better task-shared feature representations, but valuable information may be retained in task-specific decoders. In addition, each decoder imposes a burden on the number of parameters in the model.

## C    DETAILS OF CLASS-BALANCED FILTERING

Figure 8 shows the prediction process of the proposed network. The proposed method uses the coarse-grained event perception task as a class-balanced filter before event detection. First, the coarse-grained event perception task works on segment-level sequences to initially filter the input sequences that are unlikely to have events. In this way, a class-imbalanced environment with a large number of background classes is transformed into into a relatively class-balanced environment. Ideally, each input sequence in the relatively class-balanced environment contains at least one event. Next, the data in the relatively class-balanced environment is used for further fine-grained event identification.

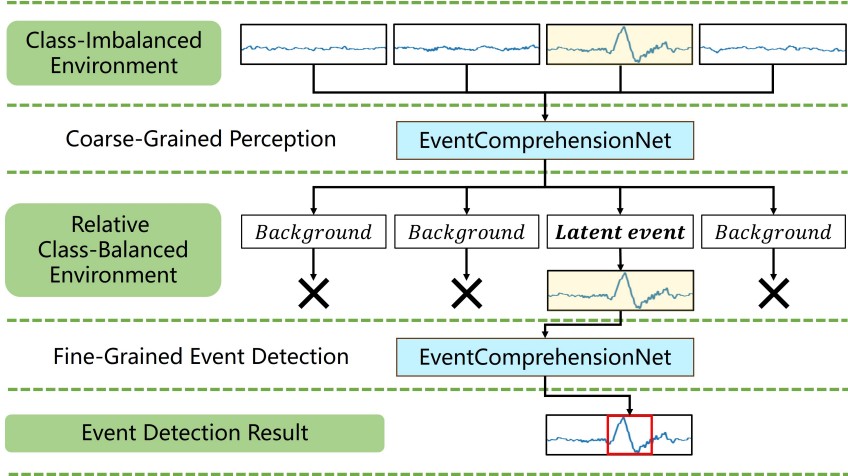

Figure 8: The prediction process of the proposed model. The coarse-grained perception task is used to filter background input sequences, providing a relatively class-balanced environment for the model.

## D    DETAILS OF EXPERIMENTAL DATASETS

**TDCS-FOG:** It collects lower-back 3D accelerometer data from subjects, and identified the precise start and end times when the subject experienced three types of freezing of gait (Start Hesitation, Turn, and Walking). It contains 833 fragments from 62 subjects (128 Hz; three axes: vertical, mediolateral, and anteroposterior).

**DREAMS:** It collects 30-minute sleep EEG data fragments from subjects which contain sleep spindle wave labels. According to You et al. (2021), the labeled EEG channel is upsampled to 256 Hz for detection.

**SHHS:** Subset SHHS-1 contains obstructive apnea (OSA) labels of $SpO_2$ signals from 5793 subjects sleeping throughout the night. We select the first 1000 subjects for OSA detection (1 Hz sampling).

**QT:** An ECG dataset with expert annotations for QRS complexes, containing diverse morphologies and high-quality labels to evaluate precise delineation.

# E  DETAILS OF BASELINE METHODS

**SpindleU-Net** (You et al., 2021): U-Net structure with attention for spindle detection and a loss to alleviate class imbalance.

**SUMO** (Kaulen et al., 2022): U-Net based, producing detection results close to expert labels with lower complexity.

**MS-TCT** (Dai et al., 2022): Video action detection model combining convolution and Transformers with multi-scale fusion; includes an auxiliary center branch.

**TimesNet** (Wu et al., 2023): Universal backbone leveraging TimesBlock to discover multi-periodicity and complex temporal variations.

**Informer** (Zhou et al., 2021): Long sequence forecasting model with strong classification ability in some settings.

**Non-Stationary Transformer (NSTransformer)** (Liu et al., 2022): Addresses over-stationarization in forecasting; effective in some classification tasks.

**TimeMixer** (Wang et al., 2024): Fully-MLP backbone with past/future mixing blocks for general time series modeling.

**QRSU-Net** (Urteaga et al., 2025): U-Net tailored for ECG QRS detection with morphological constraints and multi-scale context encoding.

# F  CALCULATION OF EVENT-LEVEL F1-SCORE

The intersection over union (IoU) of the predicted event and the ground-truth (GT) event is used to measure whether the prediction hits the GT. When multiple predictions hit the same GT, only the one with the highest IoU counts as true positive (TP). Predictions not hitting any GT are false positives (FP), and GT events not hit by any prediction are false negatives (FN). Event-level F1-score is computed as $F1 = 2 \cdot TP/(2 \cdot TP + FP + FN)$. We use IoU=0.5 by default.

# G  EFFECTIVENESS OF CLASS-BALANCED FILTERING

Figure 9 shows the confusion matrix of the coarse-grained event awareness task on the TDCS-FOG test set. 82% of background sequences are filtered out, and the remaining 18% of background sequences and 88% of event sequences are sent to the subsequent event detection process, yielding a relatively class-balanced environment.

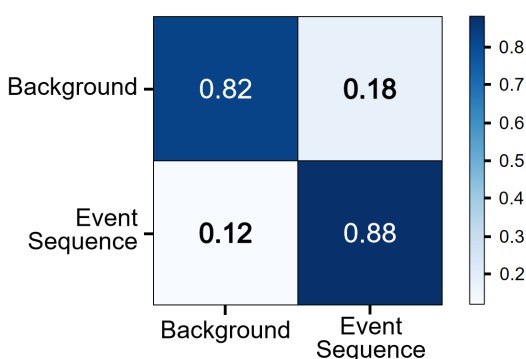

Figure 9: Confusion matrix of the coarse-grained event awareness task on TDCS-FOG (test).

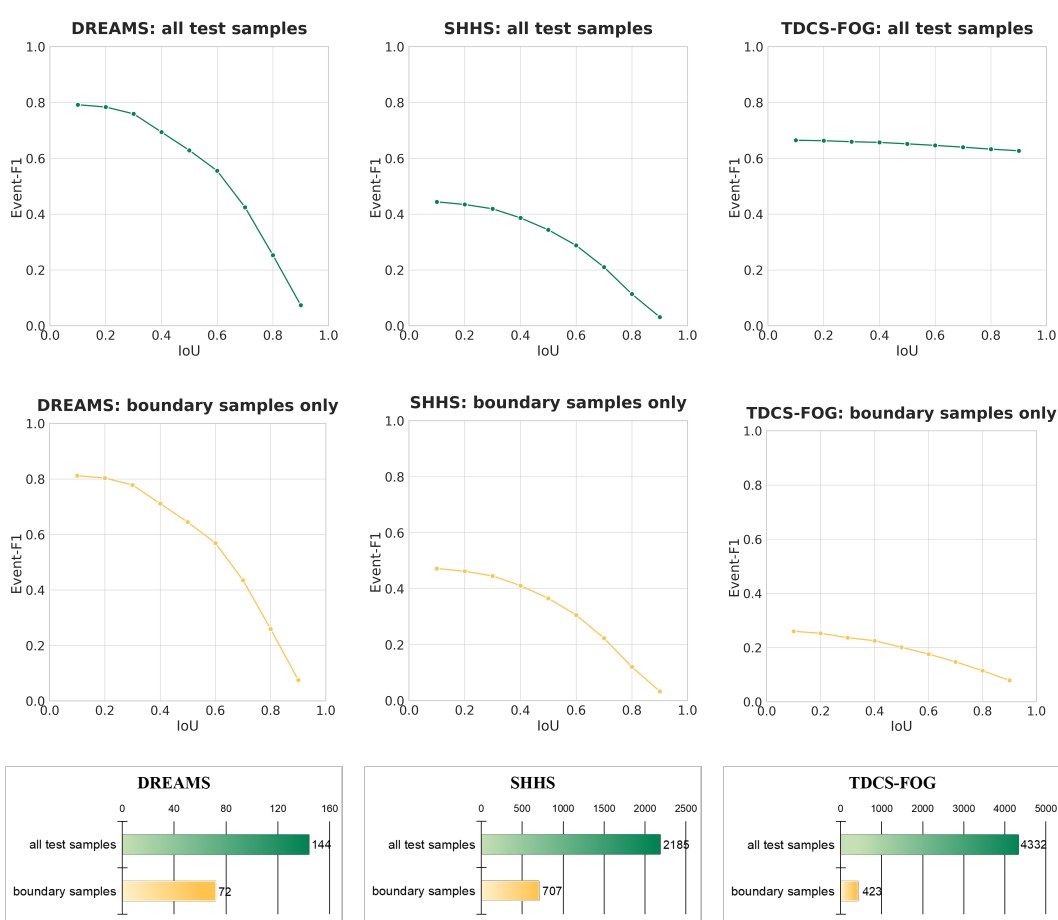

Figure 10: Trends of event-level F1-score under different IoU thresholds across datasets, and the numbers of total and boundary test samples.

## H EVENT-LEVEL DETECTION WITH DIFFERENT IOU THRESHOLDS

Figure 10 shows event-level F1 under different IoU thresholds on all samples and on boundary-only samples. A "boundary sample" contains both event and background frames within the input window, while a "non-boundary sample" contains frames of only one class. Because event durations can be

much longer than the model window on some datasets (e.g., TDCS), only a small fraction of windows are boundary samples, making boundary localization harder and more sensitive to IoU.

## I   PARAMETER ADVANTAGE OF TDC FRAMEWORK

Figure 11 shows the parameter growth comparison between the Traditional and TDC frameworks. The TDC framework maintains a constant parameter count as the number of tasks increases, effectively reducing model complexity and memory footprint compared with the traditional framework.

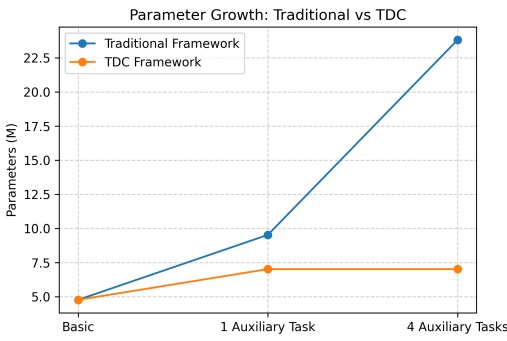

Figure 11: Parameter growth: Traditional vs TDC Framework.

## J   DIFFERENCES BETWEEN FINE-GRAINED EVENT DETECTION AND ANOMALY DETECTION

Time series anomaly detection methods often target outliers, which may not coincide with clinically meaningful events. Events of interest in this work are not necessarily outliers, and outliers may not correspond to target events. Therefore, anomaly detection approaches are not universally applicable to fine-grained event detection tasks considered here.

## K   MODEL SOURCE CODE

We provide core code in the supplementary materials (EventCompreNet_code): model building (EventCompreNet.py), data loading (load_data.py), and training/testing (train_and_test.py). The full code and baselines will be open-sourced upon acceptance.

## L   IMPLEMENTATION DETAILS AND HYPERPARAMETER SETTINGS

All models undergo class-balance pre-processing during training (minority event-containing sequences are resampled to match the majority background sequences). Each model adds a smoothing layer to smooth frame-level outputs (Kaulen et al., 2022). On SHHS, as apnea manifests on $SpO_2$ after 8–10 seconds, an input delay is used as a tunable hyperparameter.

Table 4: Best hyperparameters for EventCompreNet.

| Hyperparameter | TDCS-FOG | DREAMS | SHHS | QT |
|---|---|---|---|---|
| learning_rate | 0.001 | 0.001 | 0.0001 | 0.001 |
| batch_size | 512 | 128 | 256 | 32 |
| window_size | 320 | 3840 | 320 | 256 |
| weight_decay | 0.0002 | 0.0000 | 0.0002 | 0.0005 |
| dropout_rate | 0.0 | 0.1 | 0.0 | 0.0 |
| convchannels | 256 | 64 | 128 | 8 |
| kernel_size | 13 | 13 | 7 | 5 |
| sampling_scale | [2,2,2,2] | [10,4,2,2] | [10,4,2,2] | [10,2,2,2] |
| class_weight_0 | 1.0 | 0.3 | 0.2 | 0.1 |
| class_weight_1 | 1.0 | 0.5 | 0.8 | 1.0 |
| class_weight_2 | 2.0 | - | - | - |
| class_weight_3 | 2.5 | - | - | - |
| main_task_weight | 1.0 | 0.9 | 0.9 | 0.7 |
| center_task_weight | 0.6 | 0.6 | 0.6 | 0.1 |
| boundary_task_weight | 0.4 | 0.1 | 0.5 | 0.1 |
| lifetime_task_weight | 0.8 | 1.2 | 0.4 | 0.1 |
| coarse_task_weight | 0.5 | 17.5 | 16.5 | 10.0 |
| coarse_filter_rate | 0.4 | 0.2 | 0.3 | 0.5 |
| avg_window | 320 | 100 | 1 | 20 |
| offset | - | - | 10 | - |

## M  HYPERPARAMETER SETTINGS OF BASELINE METHODS

Table 5: Best hyperparameters for MS-TCT

| Hyperparameter | TDCS-FOG | DREAMS | SHHS | QT |
|---|---|---|---|---|
| learning_rate | 0.001 | 0.001 | 0.003 | 0.001 |
| batch_size | 256 | 32 | 256 | 32 |
| window_size | 128 | 256 | 128 | 256 |
| weight_decay | 0.0002 | 0.0004 | 0.0004 | 0.0005 |
| firstchannel | 8 | 8 | 4 | 8 |
| num_heads | 8 | 2 | 2 | 1 |
| num_block | 3 | 4 | 2 | 3 |
| class_weight_0 | 0.6 | 0.1 | 1.0 | 0.1 |
| class_weight_1 | 2.5 | 0.8 | 500.0 | 1.0 |
| avg_window | 128 | 200 | 50 | 20 |
| offset | - | - | 8 | - |

Table 6: Best hyperparameters for TimesNet

| Hyperparameter | TDCS-FOG | DREAMS | SHHS | QT |
|---|---|---|---|---|
| learning_rate | 0.001 | 0.0005 | 0.001 | 0.001 |
| batch_size | 64 | 32 | 128 | 32 |
| window_size | 64 | 256 | 240 | 256 |
| weight_decay | 0.0004 | 0.0002 | 0.0000 | 0.0005 |
| d_model | 8 | 8 | 8 | 16 |
| top_k | 13 | 13 | 5 | 5 |
| num_kernels | 2 | 3 | 3 | 7 |
| e_layers | 2 | 2 | 2 | 2 |
| class_weight_0 | 0.3 | 0.2 | 0.3 | 0.1 |
| class_weight_1 | 1.0 | 0.5 | 0.8 | 1.0 |
| avg_window | 1 | 256 | 10 | 20 |
| offset | - | - | 0 | - |

Table 7: Best hyperparameters for Informer

| Hyperparameter | TDCS-FOG | DREAMS | SHHS | QT |
|---|---|---|---|---|
| learning_rate | 0.001 | 0.003 | 0.001 | 0.001 |
| batch_size | 128 | 16 | 128 | 32 |
| window_size | 256 | 512 | 60 | 128 |
| weight_decay | 0.0002 | 0.0004 | 0.0002 | 0.0005 |
| d_model | 32 | 32 | 32 | 16 |
| n_heads | 1 | 2 | 1 | 16 |
| e_layers | 2 | 1 | 1 | 2 |
| factor | 1 | 4 | 6 | 1 |
| class_weight_0 | 0.3 | 0.1 | 0.3 | 0.1 |
| class_weight_1 | 1.0 | 0.5 | 0.8 | 1.0 |
| avg_window | 256 | 100 | 50 | 20 |
| offset | - | - | 10 | - |

Table 8: Best hyperparameters for Non-Stationary Transformer

| Hyperparameter | TDCS-FOG | DREAMS | SHHS | QT |
|---|---|---|---|---|
| learning_rate | 0.001 | 0.001 | 0.001 | 0.001 |
| batch_size | 32 | 64 | 64 | 32 |
| window_size | 256 | 256 | 120 | 256 |
| weight_decay | 0.0002 | 0.0006 | 0.0000 | 0.0005 |
| d_model | 8 | 16 | 32 | 8 |
| n_heads | 1 | 1 | 4 | 4 |
| e_layers | 2 | 2 | 1 | 2 |
| factor | 8 | 8 | 8 | 1 |
| p_hidden_dims | [16,16] | [4,4] | [8,8] | [4,4] |
| p_hidden_layers | 2 | 2 | 2 | 1 |
| class_weight_0 | 0.3 | 0.2 | 0.3 | 0.1 |
| class_weight_1 | 0.5 | 0.8 | 0.8 | 1.0 |
| avg_window | 1 | 256 | 25 | 20 |
| offset | - | - | 10 | - |

Table 9: Best hyperparameters for SpindleU-Net

| Hyperparameter | TDCS-FOG | DREAMS | SHHS | QT |
|---|---|---|---|---|
| learning_rate | 0.001 | 0.0005 | 0.003 | 0.001 |
| batch_size | 64 | 32 | 256 | 32 |
| window_size | 512 | 2048 | 960 | 1536 |
| weight_decay | 0.0002 | 0.0004 | 0.0000 | 0.0005 |
| kernel_size | 7 | 5 | 5 | 5 |
| class_weight_0 | 0.6 | 0.1 | 0.3 | 0.1 |
| class_weight_1 | 2.0 | 0.5 | 0.8 | 1.0 |
| avg_window | 500 | 100 | 25 | 7 |
| offset | - | - | 10 | - |

Table 10: Best hyperparameters for SUMO

| Hyperparameter | TDCS-FOG | DREAMS | SHHS | QT |
|---|---|---|---|---|
| learning_rate | 0.001 | 0.0005 | 0.001 | 0.001 |
| batch_size | 64 | 32 | 64 | 32 |
| window_size | 256 | 2048 | 2560 | 256 |
| weight_decay | 0.0004 | 0.0002 | 0.0002 | 0.0005 |
| convchannels | 128 | 64 | 64 | 32 |
| class_weight_0 | 0.6 | 0.2 | 0.3 | 0.1 |
| class_weight_1 | 2.0 | 2.0 | 1.0 | 1.0 |
| avg_window | 200 | 100 | 10 | 20 |
| offset | - | - | 8 | - |

Table 11: Best hyperparameters for QRSU-Net

| Hyperparameter | TDCS-FOG | DREAMS | SHHS | QT |
|---|---|---|---|---|
| learning_rate | 0.001 | 0.001 | 0.001 | 0.001 |
| batch_size | 64 | 64 | 64 | 64 |
| window_size | 256 | 2048 | 960 | 256 |
| weight_decay | 0.0002 | 0.0002 | 0.0002 | 0.0005 |
| kernel_size | 18 | 18 | 18 | 5 |
| convchannels | 24 | 24 | 24 | 24 |
| class_weight_0 | 0.6 | 0.1 | 0.3 | 0.1 |
| class_weight_1 | 2.0 | 1.0 | 0.8 | 0.8 |
| avg_window | 200 | 25 | 25 | 100 |
| offset | - | - | 8 | - |

Table 12: Best hyperparameters for TimeMixer

| Hyperparameter | TDCS-FOG | DREAMS | SHHS | QT |
|---|---|---|---|---|
| learning_rate | 0.0001 | 0.0001 | 0.0001 | 0.001 |
| batch_size | 256 | 16 | 64 | 32 |
| window_size | 256 | 512 | 256 | 256 |
| weight_decay | 0.0000 | 0.0005 | 0.0000 | 0.0005 |
| ff_dim | 64 | 512 | 64 | 256 |
| hidden_dim | 1024 | 2048 | 2048 | 1024 |
| e_layers | 2 | 6 | 2 | 2 |
| down_sampling_layers | 1 | 1 | 1 | 1 |
| down_sampling_window | 2 | 2 | 2 | 2 |
| top_k | 5 | 5 | 5 | 5 |
| class_weight_0 | 0.3 | 0.1 | 0.1 | 0.1 |
| class_weight_1 | 2.0 | 1.0 | 1.0 | 1.0 |
| avg_window | 400 | 200 | 20 | 25 |
| offset | - | - | 8 | - |

## N TUNING RANGES OF ALL MODELS

Due to space limits, we summarize representative tuning ranges below. Full ranges are consistent with the appendix and can be provided as supplementary files upon request.

Table 13: Hyperparameter tuning range of MS-TCT on TDCS-FOG

| Hyperparameter | Range |
| --- | --- |
| learning_rate | 0.003, 0.001, 0.0005, 0.0001 |
| batch_size | 64, 128, 256 |
| window_size | 128, 256, 512, 1024, 1280 |
| weight_decay | 0.0000, 0.0002, 0.0004 |
| firstchannel | 4, 8, 16 |
| num_heads | 1, 2, 4, 8 |
| num_block | 1, 2, 3, 4 |
| class_weight_0 | 0.1, 0.2, 0.4, 0.6, 0.8, 1.0 |
| class_weight_1 | 0.4, 0.8, 1.0, 2.0, 2.5, 3.0 |
| avg_window | 1, 10, 25, 50, 100, 128, 200, 500, 1000, 1280 |

Table 14: Hyperparameter tuning range of MS-TCT on DREAMS

| Hyperparameter | Range |
| --- | --- |
| learning_rate | 0.003, 0.001, 0.0005, 0.0001 |
| batch_size | 32, 64, 128, 256 |
| window_size | 256, 512, 1024, 2048 |
| weight_decay | 0.0000, 0.0002, 0.0004 |
| firstchannel | 4, 8, 16 |
| num_heads | 1, 2, 4, 8 |
| num_block | 1, 2, 3, 4 |
| class_weight_0 | 0.1, 0.2 |
| class_weight_1 | 0.5, 0.8, 1.0, 2.0 |
| avg_window | 10, 25, 50, 100, 200, 500, 1000, 2048 |

Table 15: Hyperparameter tuning range of MS-TCT on SHHS

| Hyperparameter | Range |
| --- | --- |
| learning_rate | 0.003, 0.001, 0.0005, 0.0001 |
| batch_size | 128, 256, 512 |
| window_size | 64, 128, 256, 512, 640, 960, 1280, 2048 |
| weight_decay | 0.0000, 0.0002, 0.0004, 0.001 |
| firstchannel | 16, 32, 64, 128 |
| num_heads | 1, 2, 4 |
| num_block | 2, 3, 4 |
| class_weight_0 | 0.1, 0.2, 0.3 |
| class_weight_1 | 0.5, 0.8, 1.0, 2.0, 2.5 |
| avg_window | 1, 10, 25, 50, 100, 500, 1000, 2048 |
| offset | 0, 8, 10, 12, 18 |

Table 16: Hyperparameter tuning range of MS-TCT on QT

| Hyperparameter | Range |
|---|---|
| learning_rate | 0.003, 0.001, 0.0005, 0.0001 |
| batch_size | 128, 256, 512 |
| window_size | 64, 128, 256, 512, 640, 960, 1024, 1536, 2048 |
| weight_decay | 0.0000, 0.0002, 0.0005, 0.001 |
| firstchannel | 2, 4, 8, 12, 16, 32, 64, 128 |
| num_heads | 1, 2, 4 |
| num_block | 1, 2, 3, 4 |
| class_weight_0 | 0.1, 0.2, 0.3 |
| class_weight_1 | 0.5, 0.8, 1.0, 2.0, 2.5 |
| avg_window | 1, 10, 15, 20, 25, 50, 100, 500, 1000 |

Table 17: Hyperparameter tuning range of TimesNet on TDCS-FOG

| Hyperparameter | Range |
|---|---|
| learning_rate | 0.003, 0.001, 0.0005, 0.0001 |
| batch_size | 16, 32, 64 |
| window_size | 64, 256, 1024, 1920 |
| weight_decay | 0.0000, 0.0002, 0.0004 |
| d_model | 8, 16, 32, 64 |
| top_k | 5, 9, 13 |
| num_kernels | 1, 2, 3, 4 |
| e_layers | 1, 2 |
| class_weight_0 | 0.1, 0.3 |
| class_weight_1 | 0.5, 0.7, 1.0, 2.0 |
| class_weight_2 | 0.5, 0.7, 1.0, 2.0 |
| class_weight_3 | 0.5, 0.7, 1.0, 2.0 |
| avg_window | 1, 100, 500, 1000, 1920 |

Table 18: Hyperparameter tuning range of TimesNet on DREAMS

| Hyperparameter | Range |
|---|---|
| learning_rate | 0.003, 0.001, 0.0005, 0.0001 |
| batch_size | 16, 32, 64, 128 |
| window_size | 256, 512, 768, 1024, 1280, 2560, 3840 |
| weight_decay | 0.0000, 0.0002, 0.0004, 0.0006 |
| d_model | 8, 16, 32, 64 |
| top_k | 5, 7, 9, 11, 13 |
| num_kernels | 2, 3, 4, 5 |
| e_layers | 1, 2 |
| class_weight_0 | 0.1, 0.2, 0.3 |
| class_weight_1 | 0.5, 0.8, 1.0, 2.0, 3.0 |
| avg_window | 1, 10, 100, 256, 500, 1000, 3840 |

Table 19: Hyperparameter tuning range of TimesNet on SHHS

| Hyperparameter | Range |
|---|---|
| learning_rate | 0.003, 0.001, 0.0005, 0.0001 |
| batch_size | 16, 32, 128, 256 |
| window_size | 60, 120, 240, 480, 512, 960, 1024, 2048 |
| weight_decay | 0.0000, 0.0002, 0.0004 |
| d_model | 8, 16, 32 |
| top_k | 5, 13 |
| num_kernels | 2, 3, 4 |
| e_layers | 1, 2 |
| class_weight_0 | 0.2, 0.3 |
| class_weight_1 | 0.5, 0.8, 1.0, 2.0 |
| avg_window | 1, 10, 25, 50, 100, 200, 500, 1000, 2048 |
| offset | 0, 8, 12 |

Table 20: Hyperparameter tuning range of TimesNet on QT

| Hyperparameter | Range |
|---|---|
| learning_rate | 0.003, 0.001, 0.0005, 0.0001 |
| batch_size | 16, 32, 64, 128, 256 |
| window_size | 64, 128, 256, 512, 1024, 1536, 2048 |
| weight_decay | 0.0000, 0.0002, 0.0005 |
| d_model | 8, 16, 32 |
| top_k | 5, 7, 9, 13 |
| num_kernels | 2, 3, 5, 7 |
| e_layers | 1, 2, 3 |
| class_weight_0 | 0.1, 0.2, 0.3 |
| class_weight_1 | 0.5, 0.8, 1.0, 2.0 |
| avg_window | 1, 10, 15, 20, 25, 50, 100, 200, 500, 1000 |

Table 21: Hyperparameter tuning range of Informer on TDCS-FOG

| Hyperparameter | Range |
|---|---|
| learning_rate | 0.003, 0.001, 0.0005, 0.0001 |
| batch_size | 16, 32, 64, 128 |
| window_size | 64, 256, 1024, 1920 |
| weight_decay | 0.0000, 0.0002, 0.0004 |
| d_model | 8, 16, 32, 64 |
| n_heads | 1, 2, 4 |
| e_layers | 1, 2 |
| factor | 1, 4, 8 |
| class_weight_0 | 0.1, 0.3 |
| class_weight_1 | 0.5, 0.7, 1.0, 2.0 |
| avg_window | 1, 100, 256, 500, 1000, 1920 |

Table 22: Hyperparameter tuning range of Informer on DREAMS

| Hyperparameter | Range |
|---|---|
| learning_rate | 0.003, 0.001, 0.0005 |
| batch_size | 16, 32, 64, 128 |
| window_size | 64, 128, 256, 512, 768, 1024 |
| weight_decay | 0.0001, 0.0002, 0.0004, 0.0006 |
| d_model | 32, 64 |
| n_heads | 2, 4 |
| e_layers | 1, 2 |
| factor | 1, 2, 3, 4, 5 |
| class_weight_0 | 0.1, 0.2 |
| class_weight_1 | 0.5, 0.8, 1.0, 1.5, 2.0 |
| avg_window | 1, 10, 25, 50, 100, 200, 250, 500 |

Table 23: Hyperparameter tuning range of Informer on SHHS

| Hyperparameter | Range |
|---|---|
| learning_rate | 0.001, 0.0005, 0.0001 |
| batch_size | 64, 128 |
| window_size | 60, 120, 240, 480, 512, 960 |
| weight_decay | 0.0000, 0.0002, 0.0004 |
| d_model | 8, 16, 32 |
| n_heads | 1, 2, 4 |
| e_layers | 1, 2 |
| factor | 1, 2, 6, 8 |
| class_weight_0 | 0.1, 0.2, 0.3 |
| class_weight_1 | 0.8, 1.0, 2.0 |
| avg_window | 1, 10, 25, 50, 100, 200, 960 |
| offset | 0, 10, 12, 14 |

Table 24: Hyperparameter tuning range of Informer on QT

| Hyperparameter | Range |
|---|---|
| learning_rate | 0.003, 0.001, 0.0005, 0.0001 |
| batch_size | 16, 32, 64, 128 |
| window_size | 64, 128, 256, 512, 960 |
| weight_decay | 0.0000, 0.0002, 0.0005 |
| d_model | 4, 8, 16, 32, 64 |
| n_heads | 1, 2, 4, 8, 16 |
| e_layers | 1, 2 |
| factor | 1, 2, 6, 8 |
| class_weight_0 | 0.1, 0.2, 0.3 |
| class_weight_1 | 0.8, 1.0, 2.0 |
| avg_window | 1, 10, 15, 20, 25, 50, 100, 200, 500 |

Table 25: Hyperparameter tuning range of Non-Stationary Transformer on TDCS-FOG

| Hyperparameter | Range |
|---|---|
| learning_rate | 0.003, 0.001, 0.0005, 0.0001 |
| batch_size | 16, 32 |
| window_size | 64, 256, 1024 |
| weight_decay | 0.0000, 0.0002, 0.0004 |
| d_model | 8 |
| n_heads | 1, 2 |
| e_layers | 1, 2 |
| factor | 1, 4, 8 |
| p_hidden_dims | [8,8], [16,16] |
| p_hidden_layers | 1, 2 |
| class_weight_0 | 0.1, 0.3 |
| class_weight_1 | 0.5, 0.7, 1.0, 2.0 |
| avg_window | 1, 100, 500, 1000, 1024 |

Table 26: Hyperparameter tuning range of Non-Stationary Transformer on DREAMS

| Hyperparameter | Range |
|---|---|
| learning_rate | 0.003, 0.001, 0.0005, 0.0001 |
| batch_size | 32, 64, 128 |
| window_size | 64, 128, 256, 512, 768, 1024 |
| weight_decay | 0.0000, 0.0002, 0.0004, 0.0006 |
| d_model | 8, 16, 32 |
| n_heads | 1, 2 |
| e_layers | 1, 2 |
| factor | 1, 2, 4, 6 |
| p_hidden_dims | [4,4], [8,8], [16,16] |
| p_hidden_layers | 1, 2 |
| class_weight_0 | 0.1, 0.2, 0.3 |
| class_weight_1 | 0.5, 2.0, 2.5 |
| avg_window | 1, 10, 25, 50, 100, 200, 256, 500 |

Table 27: Hyperparameter tuning range of Non-Stationary Transformer on SHHS

| Hyperparameter | Range |
|---|---|
| learning_rate | 0.001, 0.0005, 0.0001 |
| batch_size | 16, 32, 64 |
| window_size | 60, 120, 240, 480, 512, 960 |
| weight_decay | 0.0000, 0.0002, 0.0004 |
| d_model | 16, 32 |
| n_heads | 2, 4 |
| e_layers | 1, 2 |
| factor | 1, 2, 4, 8 |
| p_hidden_dims | [4,4], [8,8], [16,16] |
| p_hidden_layers | 1, 2 |
| class_weight_0 | 0.1, 0.2, 0.3 |
| class_weight_1 | 0.5, 0.8, 1.0, 3.0 |
| avg_window | 1, 10, 25, 50, 100, 200, 500, 960 |
| offset | 0, 8, 9, 10, 12 |

Table 28: Hyperparameter tuning range of Non-Stationary Transformer on QT

| Hyperparameter | Range |
|---|---|
| learning_rate | 0.003, 0.001, 0.0005, 0.0001 |
| batch_size | 16, 32, 64 |
| window_size | 64, 128, 256, 512, 1024 |
| weight_decay | 0.0000, 0.0002, 0.0005 |
| d_model | 4, 8, 16, 32 |
| n_heads | 1, 2, 4 |
| e_layers | 1, 2 |
| factor | 1, 2, 4, 8 |
| p_hidden_dims | [4,4], [8,8], [16,16] |
| p_hidden_layers | 1, 2 |
| class_weight_0 | 0.1, 0.2, 0.3 |
| class_weight_1 | 0.5, 0.8, 1.0, 3.0 |
| avg_window | 1, 10, 15, 20, 25, 50, 100, 200, 500, 1000 |

Table 29: Hyperparameter tuning range of TimeMixer on TDCS-FOG

| Hyperparameter | Range |
|---|---|
| learning_rate | 0.003, 0.001, 0.0005, 0.0001 |
| batch_size | 8, 16, 32, 64, 128, 256 |
| window_size | 64, 256, 512, 1024 |
| weight_decay | 0.0000, 0.0002, 0.0005 |
| ff_dim | 8, 32, 64, 128, 256, 512, 1024, 2048 |
| hidden_dim | 8, 32, 64, 128, 256, 512, 1024, 2048, 3072, 4096 |
| e_layers | 1, 2, 3 |
| down_sampling_layers | 1, 2 |
| down_sampling_window | 2, 4 |
| top_k | 5, 7, 9 |
| class_weight_0 | 0.1, 0.3 |
| class_weight_1 | 0.5, 0.7, 1.0, 2.0 |
| avg_window | 1, 50, 100, 200, 300, 400, 500, 1000 |

Table 30: Hyperparameter tuning range of TimeMixer on DREAMS

| Hyperparameter | Range |
|---|---|
| learning_rate | 0.003, 0.001, 0.0005, 0.0001 |
| batch_size | 8, 16, 32, 64, 128, 256 |
| window_size | 64, 256, 512, 1024 |
| weight_decay | 0.0000, 0.0002, 0.0005 |
| ff_dim | 8, 32, 64, 128, 256, 512, 1024, 2048 |
| hidden_dim | 8, 32, 64, 128, 256, 512, 1024, 2048, 3072, 4096 |
| e_layers | 1, 2, 3, 4, 5, 6 |
| down_sampling_layers | 1, 2 |
| down_sampling_window | 2, 4 |
| top_k | 5, 7, 9 |
| class_weight_0 | 0.1, 0.3 |
| class_weight_1 | 0.5, 0.7, 1.0, 2.0 |
| avg_window | 1, 50, 100, 200, 300, 400, 500, 1000 |

Table 31: Hyperparameter tuning range of TimeMixer on SHHS

| Hyperparameter | Range |
|---|---|
| learning_rate | 0.003, 0.001, 0.0005, 0.0001 |
| batch_size | 8, 16, 32, 64, 128, 256 |
| window_size | 64, 256, 512, 1024 |
| weight_decay | 0.0000, 0.0002, 0.0005 |
| ff_dim | 8, 32, 64, 128, 256, 512, 1024, 2048 |
| hidden_dim | 8, 32, 64, 128, 256, 512, 1024, 2048, 3072, 4096 |
| e_layers | 1, 2, 3 |
| down_sampling_layers | 1, 2 |
| down_sampling_window | 2, 4 |
| top_k | 5, 7, 9 |
| class_weight_0 | 0.1, 0.3 |
| class_weight_1 | 0.5, 0.7, 1.0, 2.0 |
| avg_window | 1, 10, 15, 20, 25, 50, 100, 200, 500, 1000 |
| offset | 0, 7, 8, 9, 10, 12 |

Table 32: Hyperparameter tuning range of TimeMixer on QT

| Hyperparameter | Range |
|---|---|
| learning_rate | 0.003, 0.001, 0.0005, 0.0001 |
| batch_size | 8, 16, 32, 64, 128, 256 |
| window_size | 64, 256, 512, 1024 |
| weight_decay | 0.0000, 0.0002, 0.0005 |
| ff_dim | 8, 32, 64, 128, 256, 512, 1024, 2048 |
| hidden_dim | 8, 32, 64, 128, 256, 512, 1024, 2048, 3072, 4096 |
| e_layers | 1, 2, 3 |
| down_sampling_layers | 1, 2 |
| down_sampling_window | 2, 4 |
| top_k | 5, 7, 9 |
| class_weight_0 | 0.1, 0.3 |
| class_weight_1 | 0.5, 0.7, 1.0, 2.0 |
| avg_window | 1, 10, 15, 20, 25, 50, 100, 200, 500, 1000 |

Table 33: Hyperparameter tuning range of SpindleU-Net on TDCS-FOG

| Hyperparameter | Range |
|---|---|
| learning_rate | 0.003, 0.001, 0.0005, 0.0001 |
| batch_size | 64, 128, 256, 512 |
| window_size | 128, 256, 512, 1024, 2048 |
| weight_decay | 0.0000, 0.0002, 0.0004 |
| kernel_size | 3, 5, 7, 11, 13 |
| class_weight_0 | 0.4, 0.6 |
| class_weight_1 | 0.1, 2.0 |
| class_weight_2 | 0.2, 1.0 |
| class_weight_3 | 1.0, 2.0 |
| avg_window | 1, 10, 25, 50, 100, 200, 500, 1000, 2048 |

Table 34: Hyperparameter tuning range of SpindleU-Net on DREAMS

| Hyperparameter | Range |
|---|---|
| learning_rate | 0.003, 0.001, 0.0005, 0.0001 |
| batch_size | 32, 64, 128, 256, 512 |
| window_size | 256, 512, 1024, 2048 |
| weight_decay | 0.0000, 0.0002, 0.0004 |
| kernel_size | 3, 5, 7, 11, 13 |
| class_weight_0 | 0.1, 0.2 |
| class_weight_1 | 0.5, 0.8, 1.0, 2.0 |
| avg_window | 1, 10, 25, 50, 100, 200, 500, 1000 |

Table 35: Hyperparameter tuning range of SpindleU-Net on SHHS

| Hyperparameter | Range |
|---|---|
| learning_rate | 0.003, 0.001, 0.0005, 0.0001 |
| batch_size | 64, 128, 256, 512 |
| window_size | 64, 256, 640, 960, 2048 |
| weight_decay | 0.0000, 0.0002, 0.0004 |
| kernel_size | 3, 5, 7, 11, 13 |
| class_weight_0 | 0.2, 0.3 |
| class_weight_1 | 0.5, 0.8, 1.0, 2.0 |
| avg_window | 1, 10, 25, 50, 100, 200, 500, 1000, 2048 |
| offset | 0, 8, 10, 12, 18 |

Table 36: Hyperparameter tuning range of SpindleU-Net on QT

| Hyperparameter | Range |
|---|---|
| learning_rate | 0.003, 0.001, 0.0005, 0.0001 |
| batch_size | 16, 32, 64, 128, 256, 512 |
| window_size | 64, 128, 256, 512, 1024, 1536, 2048 |
| weight_decay | 0.0000, 0.0002, 0.0005 |
| kernel_size | 3, 5, 7, 11, 13 |
| class_weight_0 | 0.1, 0.2, 0.3 |
| class_weight_1 | 0.5, 0.8, 1.0, 2.0 |
| avg_window | 1, 5, 7, 9, 25, 50, 100, 200, 500, 1000 |

Table 37: Hyperparameter tuning range of SUMO on TDCS-FOG

| Hyperparameter | Range |
|---|---|
| learning_rate | 0.003, 0.001, 0.0005, 0.0001 |
| batch_size | 64, 128, 256, 512 |
| window_size | 64, 128, 256, 512, 1024, 1920 |
| weight_decay | 0.0000, 0.0002, 0.0004 |
| convchannels | 16, 32, 64, 128 |
| class_weight_0 | 0.1, 0.2, 0.4, 0.6, 0.8, 1.0 |
| class_weight_1 | 0.1, 0.4, 0.8, 1.0, 2.0 |
| class_weight_2 | 0.1, 0.2, 0.6, 1.0, 2.0 |
| class_weight_3 | 0.1, 0.8, 1.0, 2.0 |
| avg_window | 1, 10, 50, 100, 200, 500, 1000 |

Table 38: Hyperparameter tuning range of SUMO on DREAMS

| Hyperparameter | Range |
|---|---|
| learning_rate | 0.003, 0.001, 0.0005, 0.0001 |
| batch_size | 32, 64, 128, 256, 512 |
| window_size | 256, 512, 1024, 2048 |
| weight_decay | 0.0000, 0.0002, 0.0004 |
| convchannels | 16, 32, 64, 128 |
| class_weight_0 | 0.1, 0.2 |
| class_weight_1 | 0.5, 0.8, 1.0, 2.0 |
| avg_window | 1, 100, 500, 1000 |

Table 39: Hyperparameter tuning range of SUMO on SHHS

| Hyperparameter | Range |
|---|---|
| learning_rate | 0.003, 0.001, 0.0005, 0.0001 |
| batch_size | 64, 128, 256 |
| window_size | 64, 128, 256, 480, 640, 960, 2560, 3840, 5120 |
| weight_decay | 0.0000, 0.0002, 0.0004 |
| convchannels | 32, 64, 128 |
| class_weight_0 | 0.2, 0.3 |
| class_weight_1 | 0.5, 0.8, 1.0, 2.0 |
| avg_window | 1, 10, 25, 50, 100, 200, 500, 1000 |
| offset | 0, 8, 10, 12 |

Table 40: Hyperparameter tuning range of SUMO on QT

| Hyperparameter | Range |
|---|---|
| learning_rate | 0.003, 0.001, 0.0005, 0.0001 |
| batch_size | 16, 32, 64, 128, 256 |
| window_size | 64, 128, 256, 480, 640, 960, 2560, 3840, 5120 |
| weight_decay | 0.0000, 0.0002, 0.0005 |
| convchannels | 32, 64, 128 |
| class_weight_0 | 0.1, 0.2, 0.3 |
| class_weight_1 | 0.5, 0.8, 1.0, 2.0 |
| avg_window | 1, 10, 15, 20, 25, 50, 100, 200, 500, 1000 |

Table 41: Hyperparameter tuning range of QRSU-Net on TDCS-FOG

| Hyperparameter | Range |
|---|---|
| learning_rate | 0.003, 0.001, 0.0005, 0.0001 |
| batch_size | 32, 64, 128, 256 |
| window_size | 64, 128, 256, 512, 1024, 2048 |
| weight_decay | 0.0000, 0.0002, 0.0004 |
| kernel_size | 3, 5, 7, 9, 18 |
| convchannels | 16, 24, 32, 64, 128 |
| class_weight_0 | 0.1, 0.2, 0.4, 0.6, 0.8, 1.0 |
| class_weight_1 | 0.1, 0.4, 0.8, 1.0, 2.0 |
| class_weight_2 | 0.1, 0.2, 0.6, 1.0, 2.0 |
| class_weight_3 | 0.1, 0.8, 1.0, 2.0 |
| avg_window | 1, 10, 50, 100, 200, 500, 1000 |

Table 42: Hyperparameter tuning range of QRSU-Net on DREAMS

| Hyperparameter | Range |
|---|---|
| learning_rate | 0.003, 0.001, 0.0005, 0.0001 |
| batch_size | 32, 64, 128, 256, 512 |
| window_size | 256, 512, 1024, 2048 |
| weight_decay | 0.0000, 0.0002, 0.0004 |
| kernel_size | 3, 5, 7, 9, 18 |
| convchannels | 16, 24, 32, 64, 128 |
| class_weight_0 | 0.1, 0.2 |
| class_weight_1 | 0.5, 0.8, 1.0, 2.0 |
| avg_window | 1, 25, 50, 100, 500, 1000 |

Table 43: Hyperparameter tuning range of QRSU-Net on SHHS

| Hyperparameter | Range |
|---|---|
| learning_rate | 0.003, 0.001, 0.0005, 0.0001 |
| batch_size | 32, 64, 128, 256 |
| window_size | 64, 128, 256, 480, 640, 960, 2560, 3840, 5120 |
| weight_decay | 0.0000, 0.0002, 0.0004 |
| kernel_size | 3, 5, 7, 9, 18 |
| convchannels | 16, 24, 32, 64, 128 |
| class_weight_0 | 0.1, 0.2, 0.3 |
| class_weight_1 | 0.5, 0.8, 1.0, 2.0 |
| avg_window | 1, 10, 25, 50, 100, 200, 500, 1000 |
| offset | 0, 8, 10, 12 |

Table 44: Hyperparameter tuning range of QRSU-Net on QT

| Hyperparameter | Range |
|---|---|
| learning_rate | 0.003, 0.001, 0.0005, 0.0001 |
| batch_size | 32, 64, 128, 256 |
| window_size | 64, 128, 256, 480, 640, 960, 2560, 3840, 5120 |
| weight_decay | 0.0000, 0.0002, 0.0005 |
| kernel_size | 3, 5, 7, 9, 18 |
| convchannels | 16, 24, 32, 48, 64, 128 |
| class_weight_0 | 0.1, 0.2, 0.3 |
| class_weight_1 | 0.5, 0.8, 1.0, 2.0 |
| avg_window | 1, 10, 25, 50, 100, 200, 500, 1000 |

Table 45: Hyperparameter tuning range of EventCompreNet on TDCS-FOG

| Hyperparameter | Range |
|---|---|
| learning_rate | 0.003, 0.001, 0.0005, 0.0001 |
| batch_size | 64, 128, 256, 512 |
| window_size | 320, 640, 960, 1280, 1600, 1920 |
| weight_decay | 0.0000, 0.0002, 0.0004 |
| dropout_rate | 0.0, 0.1, 0.2, 0.3, 0.4, 0.5 |
| convchannels | 64, 128, 256 |
| kernel_size | 7, 9, 13, 15 |
| sampling_scale | [2, 2, 2, 2], [10, 4, 2, 2], [8, 5, 2, 2], [8, 5, 4, 2] |
| class_weight_0 | 0.1, 0.2, 0.4, 0.6, 0.8, 1.0 |
| class_weight_1 | 0.4, 0.8, 1.0, 2.0, 2.5, 3.0 |
| class_weight_2 | 0.2, 0.6, 1.0, 2.0, 2.5, 3.0 |
| class_weight_3 | 0.8, 1.0, 2.0, 2.5, 3.0 |
| main_task_weight | 0.3, 0.9, 1.0, 1.2 |
| center_task_weight | 0.4, 0.6, 0.8, 1.0, 1.2 |
| boundary_task_weight | 0.1, 0.2, 0.3, 0.4 |
| lifetime_task_weight | 0.2, 0.8, 1.0 |
| coarse_task_weight | 0.5, 16.5, 17.5 |
| coarse_filter_rate | 0.1, 0.2, 0.3, 0.4 |
| avg_window | 10, 100, 200, 320, 500, 1000, 1920 |

Table 46: Hyperparameter tuning range of EventCompreNet on DREAMS

| Hyperparameter | Range |
|---|---|
| learning_rate | 0.003, 0.001, 0.0005, 0.0001 |
| batch_size | 32, 64, 128, 256 |
| window_size | 320, 640, 1280, 2560, 3840 |
| weight_decay | 0.0000, 0.0002, 0.0004 |
| dropout_rate | 0.0, 0.1, 0.2, 0.3, 0.4, 0.5 |
| convchannels | 16, 32, 64, 128 |
| kernel_size | 7, 11, 13, 15 |
| sampling_scale | [2, 2, 2, 2], [10, 4, 2, 2], [8, 5, 2, 2], [8, 5, 4, 2] |
| class_weight_0 | 0.1, 0.2, 0.3 |
| class_weight_1 | 0.5, 0.8, 1.0, 2.0, 3.0 |
| main_task_weight | 0.3, 0.5, 0.7, 0.9, 1.0, 1.2 |
| center_task_weight | 0.2, 0.4, 0.6, 0.8, 1.0, 1.2 |
| boundary_task_weight | 0.1, 0.3, 0.5, 0.7 |
| lifetime_task_weight | 0.4, 0.6, 0.8, 1.0, 1.2 |
| coarse_task_weight | 0.3, 0.5, 16.5, 17.5 |
| coarse_filter_rate | 0.1, 0.2, 0.3, 0.4 |
| avg_window | 1, 100, 500, 1000, 3840 |

Table 47: Hyperparameter tuning range of EventCompreNet on SHHS

| Hyperparameter | Range |
|---|---|
| learning_rate | 0.003, 0.001, 0.0005, 0.0001 |
| batch_size | 32, 64, 128, 256, 512 |
| window_size | 160, 320, 640, 960, 1280, 2560, 3840 |
| weight_decay | 0.0000, 0.0002, 0.0004 |
| dropout_rate | 0.0, 0.1, 0.2, 0.3, 0.4, 0.5 |
| convchannels | 32, 64, 128 |
| kernel_size | 7, 11, 15 |
| sampling_scale | [2, 2, 2, 2], [10, 4, 2, 2], [8, 5, 2, 2], [8, 5, 4, 2] |
| class_weight_0 | 0.1, 0.2 |
| class_weight_1 | 0.5, 0.8, 1.0 |
| main_task_weight | 0.7, 0.9, 1.0, 1.2 |
| center_task_weight | 0.4, 0.6, 0.8, 1.0, 1.2 |
| boundary_task_weight | 0.5, 0.7, 0.9 |
| lifetime_task_weight | 0.2, 0.4, 0.6, 0.8, 1.0 |
| coarse_task_weight | 0.5, 16.5, 17.5 |
| coarse_filter_rate | 0.1, 0.2, 0.3, 0.4 |
| avg_window | 1, 10, 25, 50, 100, 200, 500, 1000, 3840 |
| offset | 0, 8, 10, 12, 18 |

Table 48: Hyperparameter tuning range of EventCompreNet on QT

| Hyperparameter | Range |
|---|---|
| learning_rate | 0.003, 0.001, 0.0005, 0.0001 |
| batch_size | 16, 32, 64, 128, 256, 512 |
| window_size | 64, 128, 256, 512, 1024, 1536, 2048, 3840 |
| weight_decay | 0.0000, 0.0002, 0.0004 |
| dropout_rate | 0.0, 0.1, 0.2, 0.3, 0.4, 0.5 |
| convchannels | 8, 16, 32, 64, 128 |
| kernel_size | 3, 5, 7, 11, 15 |
| sampling_scale | [2, 2, 2, 2], [10, 4, 2, 2], [10, 2, 2, 2] [8, 5, 2, 2], [8, 5, 4, 2] |
| class_weight_0 | 0.1, 0.2 |
| class_weight_1 | 0.5, 0.8, 1.0 |
| main_task_weight | 0.7, 0.9, 1.0, 1.2 |
| center_task_weight | 0.1, 0.4, 0.6, 0.8, 1.0, 1.2 |
| boundary_task_weight | 0.1, 0.5, 0.7, 0.9 |
| lifetime_task_weight | 0.1, 0.2, 0.4, 0.6, 0.8, 1.0 |
| coarse_task_weight | 0.5, 10.0, 15.5, 16.5, 17.5 |
| coarse_filter_rate | 0.1, 0.2, 0.3, 0.4, 0.5 |
| avg_window | 1, 10, 15, 20, 25, 50, 100, 200, 500, 1000 |

## O    USE OF LARGE LANGUAGE MODELS

Large language models (LLMs) are used solely for language editing and clarity improvement.

Purpose: We use OpenAI's ChatGPT (GPT-5, released 2025) only to polish grammar, refine wording, and improve readability of drafts.

Scope: No model output is included verbatim as research content. All conceptual contributions, analyses, data interpretation, and conclusions are entirely our own.

Verification: We review and, when necessary, revise every LLM-suggested edit to ensure factual accuracy and compliance with ethical publishing standards.

