# OpenReview forum: "Understanding Medical Time Series Event Piece by Piece: A Fine-Grained Event Detection Network"
_ICLR.cc/2026/Conference — ICLR 2026 Conference Withdrawn Submission_

### Official Review · Reviewer_JMV5 · 2025-10-29

**Soundness:** 1
**Presentation:** 3
**Contribution:** 2
**Rating:** 2
**Confidence:** 3

**Summary:**

This paper proposes EventCompareNet, a convolutional network for identifying events in medical time series.

**Strengths:**

- The extensive hyperparameter tuning of the baselines is well done
- Good selection of baselines
- Cross validation throughout the entire experimental setup helps resolve some of the concerns about the tiny dataset size

**Weaknesses:**

- If I understand the experimental setup correctly, data from a single patient might show up in both the training and test set? If so, this is a severe soundness violation.
- Reporting only F1 score is problematic. It's best to do evaluations on a couple of metrics. One particularly important missing metric is AUROC, which is threshold independent.
- The sizes of the datasets used are relatively small.
- The novelty of the task decoupling network is unclear. Task embeddings have been explored many times in prior work. See https://aclanthology.org/C18-1251.pdf for a good example
- This is a relatively minor point, but to me it's very unclear why the additional "human comprehension tasks" would help, since they are all a subset of the fine-grained event detection task.

**Questions:**

See above

---

> ### Author Response · Authors · 2025-12-03
>
> **1.  The Correct Data Splitting Strategy in the Experimental Setup**
>
> Data from the same subject do not appear in both the training and test sets. To eliminate any confusion, we define our data partition strategy as follows:
>
> - We first grouped all distinct subjects into non-overlapping "data slices" (partitions). A "fold" refers to one round of the cross-validation experiment using these pre-defined subject slices.
>
> - We performed the data splitting at the subject level before any model training. Specifically, in our k-fold cross-validation, the subject-wise data slices serve as the basic units. In each round (fold), one specific slice is designated as the test set, while the other slices form the training and validation sets.
>
> - Following this logic, in each fold, 20% of the data (constituting the test slice) is the test set, and the remaining 80% of the data is divided into the training and validation sets by a ratio of 8:2.
>
> **2. The Choice of Metrics**
>
> We agree with the reviewer’s suggestion and will consider adding threshold-independent metrics in the main text.
>
> In fact, in our original experiments we also computed **mAP (mean Average Precision)**, which is defined as the area under the Precision–Recall curve (AP) and then averaged over all classes or event types. This metric simultaneously reflects both localization accuracy and classification reliability. Due to space limitations, we previously reported only one point-wise metric and one event-level metric in the main text. We will consider adding additional metrics in the revised version.
>
> **3. Limited Dataset Size**
>
> The acquisition of fine-grained medical event data is inherently **high-cost, highly specialized, and limited in scale**. Specifically:
>
> - Data collection relies on clinical-grade multi-channel physiological equipment, which involves long experimental periods, high subject recruitment efforts, and strict ethical approvals.
> - Event-level annotations must be performed frame by frame by qualified medical experts, making the labeling process labor-intensive and challenging in terms of inter-rater consistency.
> - These events are intrinsically low-occurrence, small-sample targets within physiological signals, further exacerbating the scarcity of data.
>
> Therefore, the “small dataset” characteristic in fine-grained medical event detection is a long-standing and unavoidable constraint. To mitigate the risks associated with limited data, we adopt the following strategies:
>
> - Evaluation across multiple datasets (four public datasets covering different event types).
> - Fair comparison with multiple strong baselines under the same data splits.
> - Validation using multiple metrics, including both point-level and event-level measurements.
>
> **4. On the Prior Work of Task Embeddings**
>
> We thank the reviewer for the relevant references. The key differences can be summarized as follows:
>
> - Prior work mainly targets natural language tasks;
> - It focuses on semantic transfer and lexical-level sharing;
> - The output space is typically a discrete symbolic space (tokens).
>
> In contrast, our setting involves:
>
> - Continuous-time signals;
> - Nanosecond-to-second temporal resolution;
> - Extreme class imbalance;
> - Strong temporal structural constraints;
> - A multi-task semantic span from point → segment → cycle.
>
> Unlike NLP embeddings that aim to bridge semantic gaps across similar text domains, our task-decoupling network is designed to alleviate negative transfer caused by conflicting physiological features. By projecting shared features into task-specific subspaces, joint optimization strengthens the global representation while preserving fine-grained task-specific details. Therefore, although both paradigms formally use task embeddings, their objectives, optimization constraints, and robustness challenges are fundamentally different.
>
> **5. Why “Human Understanding Tasks” Help**
>
> From an information-theoretic perspective, if point-wise labels for fine-grained detection are already known, the HC task labels can be directly derived. Thus, they form a logical subset and introduce no new external information. However, these tasks are designed not to add information, but to **provide different optimization perspectives**.
>
> - Microscopic perspective (main task): Fine-grained detection is point-wise and local. The model may overfit high-frequency noise or memorize local waveform patterns.
> - Macroscopic perspective (auxiliary task): The human-understanding task is semantic and global. It forces the model to capture the contextual structure and overall semantics of the signal before answering questions such as “what is this waveform?” or “how many events occurred?”
>
> These two perspectives provide orthogonal gradient signals. The auxiliary tasks act as a form of regularization, preventing overfitting to local noise and encouraging the learning of robust, globally consistent features. They can therefore be viewed as feature-level data augmentation.

---

### Official Review · Reviewer_onTg · 2025-10-31

**Soundness:** 2
**Presentation:** 3
**Contribution:** 2
**Rating:** 4
**Confidence:** 4

**Summary:**

The paper proposes EventCompreNet, a multi-task learning model for fine-grained event detection in medical time series. The authors evaluate the method on four public datasets (FOG, DREAMS, SHHS, QT) and report state-of-the-art results compared with both task-specific and general-purpose time-series models.

**Strengths:**

1. The proposed TDC framework elegantly addresses a real issue in multi-task setups: limited information sharing and redundant parameters. The use of a shared decoder with task embeddings is clean and efficient.
2. The ablation study nicely isolates the effects of HC tasks and the TDC framework. Visualizations help interpret what each auxiliary task contributes.

**Weaknesses:**

1. The core idea—sharing parameters across multiple auxiliary tasks with task identifiers—has been explored in multi-task learning for years. The paper does not clearly explain what is fundamentally new beyond reusing an existing framework under a new name. To make the contribution more convincing, the authors could formalize the “task-deep-coupling” mechanism, quantify the degree of knowledge transfer, or demonstrate a measurable improvement in training dynamics over conventional shared-encoder designs.
2. The proposed auxiliary tasks are described as mimicking human understanding of events, but this analogy remains superficial. There is no cognitive evidence or ablation showing that these particular four tasks are necessary or complementary. The paper would be stronger if it replaced this speculative framing with a more principled rationale—e.g., decomposing event detection into sub-tasks based on temporal reasoning or uncertainty estimation rather than on loosely defined human cognition.
3. The “class-balanced filtering” step appears to use label information to preselect sequences containing events, which risks data leakage. In addition, some datasets are extremely small, and there is no mention of subject-wise splits or statistical significance testing

**Questions:**

1. Could the authors clarify how the proposed “task-as-sample” training scheme differs from standard multi-task batching? Does it actually expand the dataset size or simply replicate samples with different task IDs? How does it affect convergence speed and sample balance?
2. In the “class-balanced filtering” step, how does the model decide which sequences contain events without using ground-truth labels? If this decision relies on supervision, wouldn’t it lead to leakage during training or biased evaluation?

---

> ### Author Response · Authors · 2025-12-03
>
> **1. On the Novelty of “Task Deep Coupling (TDC)”**
>
> We agree that “parameter sharing + task embedding” itself is not new. Our novelty lies in the following system-level contributions:
>
> - Deep Coupling Instead of Shallow Sharing: All tasks share the same decoding pathway, feature flow, and parameters, with task semantics modulated only by task-ID embeddings—rather than using multi-head decoders.
>
> - Task-as-Sample Training Paradigm: We explicitly map the task dimension into the sample dimension, transforming multi-task optimization into conditional learning over a unified sample distribution.
>
> - First Systematic Validation in Fine-Grained Medical Event Detection: Prior task-embedding works focus mainly on NLP or recommendation systems. Our work validates its stability and generalization in high-resolution, noisy, and extremely imbalanced medical time-series data.
>
> **2. The Design of Auxiliary Tasks**
>
> Our design is inspired not by casual human perception, but by **clinical workflows of PSG interpretation**:
>
> - Coarse-grained perception: Physicians scan the night’s recordings to filter background and focus on likely event segments—corresponding to our coarse-grained perception task, similar to discarding background frames in video action localization.
>
> - Event localization (Center & Boundary): In these segments, physicians identify the core region of an event (analogous to the action climax). The center task infers the event extent from the center outward. The boundary task determines start/end times, complementing the center task by refining event boundaries “outside in.”
>
> - Event integrity judgment (Lifetime): Physicians also assess event completeness, corresponding to our lifetime task, which models event continuity.
>
> At the theoretical level, these four types of tasks constitute a *multi-scale structured decomposition* of the temporal event detection problem. They introduce interpretable mathematical constraints at the point level, interval level, and global level, respectively, thereby decomposing a highly non-convex sequence detection problem into several structurally meaningful and mathematically interpretable sub-constraints.
>
> | Task Type                       | Theoretical Role                          |
> | ------------------------------- | ----------------------------------------- |
> | Center localization             | Symmetry and center-invariance constraint |
> | Boundary localization           | Boundary separability constraint          |
> | Lifetime analysis               | Temporal scale consistency                |
> | Coarse-grained event perception | Global semantic existence constraint      |
>
> These four types of constraints form a hierarchical temporal reasoning structure, spanning from local to global, point-level to interval-level, and instantaneous to periodic scales. Therefore, they are not arbitrarily constructed “cognitive analogies,” but rather a *structural decomposition* of the event detection problem.
>
> **3. Class-Balanced Filtering**
>
> Class-balanced filtering is applied:
>
> - Only on the training set;
> - Never on validation or test sets;
> - Offline before data enter the model.
>
> Thus, it does not affect inference nor cause data leakage.
>
> **4. The Limited Size of Single Dataset**
>
> We fully acknowledge the reviewer’s valid concern regarding the limited data scale. However, it should be noted that in the field of physiological signals and fine-grained event detection—especially for tasks requiring point-wise annotations—high-quality labeled data are intrinsically extremely scarce. As a result, individual datasets are generally small in size, which is a long-standing and unavoidable constraint of this research area. We mitigate this via:
>
> - Multi-dataset evaluation;
> - Fair comparison with strong baselines;
> - Multiple complementary metrics.
>
> **5. “Task-as-Sample” vs. Standard Multi-Task Batching**
>
> - In standard multi-task learning, each sample appears at most once per task per batch.
> - In “task-as-sample,” the same window *xᵢ* is expanded as:  *(xᵢ, task₁), (xᵢ, task₂), …*
>
> Thus, the effective sample size becomes N×K in the optimization space, strengthening inter-task coupling and stabilizing shared representations via denser gradients.
>
> **6. Whether Class-Balanced Filtering Uses Ground Truth**
>
> Class-balanced filtering is applied prior to the training stage and indeed makes use of ground-truth labels; however, it does not cause data leakage during either training or evaluation. First, it is performed exclusively on the training set and is never applied to the validation or test sets. Second, the filtering is conducted offline before the data are fed into the model and does not participate in the optimization process itself; hence, from the model’s perspective, this operation is effectively “invisible”.

---

### Official Review · Reviewer_ajg8 · 2025-10-31

**Soundness:** 3
**Presentation:** 2
**Contribution:** 3
**Rating:** 4
**Confidence:** 4

**Summary:**

This paper addresses the detection of medical events from time series data, such as EEG data, where events are detected in a fine-grained manner, i.e., at the time point level. The authors propose training the main fine-grained detection model, accompanied by multi-perspective human comprehension task models, to improve detection performance. The multi-perspective human comprehension tasks consist of the center localization, boundary localization, lifetime analysis, and coarse-grained event perception tasks, where each task boosts the main fine-grained detection model, and the coarse-grained event perception acts as a class-balanced filter. They utilize a single model, EventCompreNet, which contains a task-deep-coupling framework, for all the above tasks, where the output is switched based on the task ID embedding. The proposed method was evaluated on four fine-grained event detection datasets and consistently outperformed baselines.

**Strengths:**

-- The multi-perspective human comprehension tasks and task-deep-coupling framework are intuitively reasonable to improve the fine-grained detection model. It may be related to curricuram learning, such as https://arxiv.org/abs/2212.03597 and https://www.google.com/url?sa=t&source=web&rct=j&opi=89978449&url=https://www.sciencedirect.com/science/article/pii/S0020025523003183&ved=2ahUKEwi28eKdgM6QAxV2mq8BHcZrHcYQFnoECAsQAQ&usg=AOvVaw2WbyBn9PZOoAGHBkjx9kQd.

-- Experimental results on multiple datasets demonstrated the effectiveness of the proposed method.

**Weaknesses:**

-- The proposed method may have novel points and have some practical impact, but the clarity issues are severe in understanding the method:
* The important part of the method description, "Task Synchronous Training," is hard to follow and misses a lot of details. How can we handle the differences in output variable types, including binary, multi-class, and real-valued, without modifying the network architecture, using only the task ID embeddings?
* Section "4.3 CLASS-BALANCED FILTERING" requires more details. If we naively do "The sample size for each task is repeatedly sampled to align with the largest sample size task", all the samples are classified as "background" to minimize loss.
* The descriptions for "Center Localization Task" and "Boundary Localization Task" from l.197 are hard to follow. The ideas are simple, so I can understand what is conducted in the tasks briefly. However, it is difficult to understand precisely what the labels are for the tasks.
* Eq.10 and the following description would contain a typo or be confusing.
* Eq.11 can be improved by making the second low "larger than zero" for generalizing the number of classes is larger than 1.
* In. l.126 and l.298, k is used in different meanings.
* It is better to put the tables and figures on top.

**Questions:**

-- The authors mentioned "it is the first universal network in this research area". Are there any related works in other domains?

-- Curriculum learning can be related, such as https://arxiv.org/abs/2212.03597 and https://www.google.com/url?sa=t&source=web&rct=j&opi=89978449&url=https://www.sciencedirect.com/science/article/pii/S0020025523003183&ved=2ahUKEwi28eKdgM6QAxV2mq8BHcZrHcYQFnoECAsQAQ&usg=AOvVaw2WbyBn9PZOoAGHBkjx9kQd. Is it possible to discuss with the literature?

---

> ### Author Response · Authors · 2025-12-03
>
> **1. Task-Synchronous Training and Unified Output**
>
> Our core idea is: a unified decoder architecture + task conditioning via Task-ID embeddings before the output head, rather than designing separate output branches for each task.
>
> Specifically:
>
> - Unified Latent Space: All tasks map to a shared continuous latent vector z ∈ ℝᵈ, serving as a task-agnostic “universal event representation” before conditioning.
>
> - Task-ID Embedding: For task *k*, its ID maps to an embedding eₖ, fused with encoded features for decoding. Different task semantics are realized within the same decoder without altering network topology.
>
> - Different Output Types Are Distinguished Only at the Loss Layer, Not the Network Structure:
>
>   a. Binary→ Sigmoid + BCE
>
>   b. Multi-class→ Softmax + Cross-Entropy
>
>   c. Regression → Linear + L1/L2
>
> Thus, task differences are handled at the output interpretation and loss level, enabling task-synchronous trainingwithout extra task-specific heads.
>
> **2. Class-Balanced Filtering**
>
> We appreciate the reviewer’s concern that naive resampling to the size of the largest task may cause background samples to dominate, encouraging the model to predict everything as background.
>
> We clarify that this collapse is **effectively avoided** in our setting for the following reasons:
>
> - Oversampling + Loss Re-weighting: Event-window-oriented oversampling is combined with class-balanced loss, amplifying gradients from rare events and preventing trivial all-background predictions.
> - Targeted Data Augmentation: Segments with valid events are oversampled to reinforce event-specific feature learning.
> - Empirical Validation: Experiments confirm effective boundary learning without collapsing to background-only outputs.
>
> **3. The Label Definitions of Center Localization and Boundary Localization**
>
> To clarify the label generation process, we kindly refer the reviewer to **Figure 3**, which visualizes the target signals. Specifically:
>
> - Center Localization Task:  Instead of a simple binary classification (presence/absence), we formulate this as a dense regression task. We construct a 1D Gaussian heatmap centered at the midpoint of each event. This "soft" label encourages the model to output higher probabilities as it approaches the event center, effectively guiding the model to focus on the core region.
> - Boundary Localization Task:  The "ideal" boundary labels (binary 0/1 spikes at start/end frames) are extremely sparse and difficult for deep networks to learn directly. Therefore, we adopt a similar Gaussian smoothing strategy. We generate two narrower Gaussian heatmaps centered at the start and end timestamps, respectively. This converts the task from a hard sparse detection into a soft boundary localization task, allowing the model to progressively learn the "harbingers" of event onset and termination.
>
> **4. On Extending Eq. (11) to Multi-Class Cases**
>
> We thank the reviewer for the suggestion. We intentionally adopted a **binary classification view** at the beginning of this section (together with Eq. (1)) to simplify the derivation and highlight the core filtering logic. Under this assumption, Eq. (11) naturally follows binary notation. In practice, our experiments already include multi-class scenarios, and the formulation can be straightforwardly extended as suggested.
>
> **5. On the Claim of “The First Universal Network in This Research Area”**
>
> Here, “universal network” is used specifically in the context of fine-grained medical time series event detection. To our knowledge, there exists no prior framework capable of handling multiple fine-grained medical event detection tasks within a single unified architecture.
>
> We also acknowledge that in other domains (e.g., video action detection), event modeling methods with a certain degree of “universality” already exist. To address the lack of research in this technical direction within the medical time series domain and to enable fair cross-domain comparison, we explicitly include a representative video action (event) detection model as a baseline in our experiments (I.356–I.358).
>
> **6. Relationship to Curriculum Learning (CL)**
>
> The mechanisms of CL and our method are fundamentally different.
>
> CL controls training by gradually introducing samples from easy to hard, dynamically adjusting the training sequence. Reference 1 does so via down-sampling easy samples, while Reference 2 adjusts signal smoothness across stages—both regulate difficulty at the data level, targeting the training process itself.
>
> In contrast:
>
> - All tasks are trained synchronously from the first epoch in our method—no staged task introduction.
> - The difficulty arises from task-level semantic hierarchy, not from sample hardness.
> - CL controls how samples enter training, whereas our method controls how multiple semantic views of the same sample jointly constrain optimization.

---

### Note · Authors · 2025-12-03

**Comment:**

We introduce EventCompreNet, which is the first universal framework designed for the underexplored field of **fine-grained medical time-series event detection**. By integrating human-comprehension-inspired auxiliary tasks and Task-Deep-Coupling (TDC) mechanism, our model achieves efficient knowledge transfer without parameter growth. Experiments on four public datasets demonstrate state-of-the-art performance, consistent generalization, and visualization-based interpretability, highlighting the practical applicability.

However, due to space limitations, some details were placed in the appendix, which may have caused misunderstandings. We still sincerely appreciate the reviewers' feedback.

Therefore, we have decided to withdraw this submission in order to restructure the manuscript and integrate its key content into the main text.

**Withdrawal Confirmation:**

I have read and agree with the venue's withdrawal policy on behalf of myself and my co-authors.